# Integrating leiomyoma genetics, epigenomics, and single-cell transcriptomics reveals causal genetic variants, genes, and cell types

Kadir Buyukcelebi [1,2], Alexander J. Duval[1,2], Fatih Abdula [1,2], Hoda Elkafas [1], Fidan Seker-Polat[1] & Mazhar Adli [1] ✉

Uterine fibroids (UF), that can disrupt normal uterine function and cause significant physical and psychological health problems, are observed in nearly 70% of women of reproductive age. Although heritable genetics is a significant risk factor, specific genetic variations and gene targets causally associated with UF are poorly understood. Here, we performed a meta-analysis on existing fibroid genome-wide association studies (GWAS) and integrated the identified risk loci and potentially causal single nucleotide polymorphisms (SNPs) with epigenomics, transcriptomics, 3D chromatin organization from diverse cell types as well as primary UF patient's samples. This integrative analysis identifies 24 UF-associated risk loci that potentially target 394 genes, of which 168 are differentially expressed in UF tumors. Critically, integrating this data with single-cell gene expression data from UF patients reveals the causal cell types with aberrant expression of these target genes. Lastly, CRISPR-based epigenetic repression (dCas9-KRAB) or activation (dCas9-p300) in a UF disease-relevant cell type further refines and narrows down the potential gene targets. Our findings and the methodological approach indicate the effectiveness of integrating multi-omics data with locus-specific epigenetic editing approaches for identifying gene- and celt type-targets of disease-relevant risk loci.

Uterine fibroids (UF), also called leiomyomas, are the most widely observed tumors in women of reproductive age. By 50 years of age, more than 70% of all women (70 % white and > 80% black) will develop at least one fibroid tumor. In the majority of cases, leiomyomas are non-malignant and asymptomatic; however, in 20 to 25% of cases, UF tumors can disrupt normal uterine function, resulting in a wide range of physical and health problems, including excessive uterine bleeding, anemia, defective embryo implantation, recurrent pregnancy loss, preterm labor and obstruction of labor[1]. Few medical treatments are available for UF, and many patients undergo extreme measures such as surgical hysterectomy. However, such procedures create significant

emotional stress on patients and a substantial financial burden on society. These practices cost between $5.9 billion and 34.4 billion USD in the United States alone[2].

The biological mechanisms leading to the development of leiomyomas are incompletely understood. Several risk factors, including advanced age, high body mass index, nulliparity, and the serum levels of estradiol and progesterone hormones, have been associated with the number and severity of leiomyomas[3]. At the genomic level, these tumors have a relatively low overall mutational burden compared to malignant cancers[1,4]. Whole-exome and whole-genome sequencing efforts indicate that nearly 70% of these tumors have recurrent somatic

[1]Department of Obstetrics and Gynecology, Robert Lurie Comprehensive Cancer Center, Feinberg School of Medicine at Northwestern University, Chicago, IL, USA. [2]These authors contributed equally: Kadir Buyukcelebi, Alexander J. Duval, Fatih Abdula. ✉e-mail: adli@northwestern.edu

mutations in a single gene that encodes the mediator of transcription subunit 12 (MED12) protein[4]. Furthermore, recurrent alterations in fumarate hydratase (*FH*), translocation in high mobility group AT-hook 1 and 2 (*HMGA1 and HMGA*2) genes, and deletion of collagen (*COL4A5-COL4A6* genes) are among the most frequently observed genetic alteration[5]. More recently, inactivating mutations in members of the SRCAP complex, which result in H2A.Z loading defects, were also reported[6]. Notably, the majority of these mutations are mutually exclusive, and integrative data analysis reveals mutation-specific distinct driver gene expression programs and biomarkers in UF[6,7]. Importantly, in addition to the recurrent somatic alterations, population-level studies indicate the presence of yet-to-be-clearly identified heritable genetic factors that may cause the development of leiomyoma. For example, the race and genetic background of the individual are among the most significant risk factors[8]. Additionally, the family members[9] and first-degree patient relatives have 2.5-fold greater risk of developing leiomyoma compared to the population average[10] and the concordance among monozygotic twins is almost twice that of dizygotic twins[11].

Despite this compelling evidence, identifying specific genetic variants associated with or causal to UF development is a non-trivial task. To this end, genome-wide association studies (GWAS) in a large number of affected vs. non-affected individuals have been a powerful approach. Several GWAS have been performed to identify the genetic architecture of UF risk[12–17] and have revealed numerous UF-associated SNPs and risk loci. Although findings from these studies have significantly expanded the number of identified heritable genetic risk factors, several challenges remain. First, these GWAS have identified shared and unique lead SNPs and risk loci sets. Whether the differences between studies are due to sample size or the study population's genetic ancestry is yet to be understood. Second, identifying the gene targets of lead SNPs remains challenging because greater than 90% of disease-associated SNPs are found in non-coding or intergenic genome regions[18]. Thus, they rarely create a direct alteration in a protein product[12,19]. Third, due to linkage disequilibrium (LD), GWAS typically reveals "risk loci" that contain the lead SNPs together with hundreds of other significantly associated SNPs that may span dozens of genes. Lastly, since GWAS samples are typically gathered from blood, it is impossible to identify which cell types are transcriptionally affected or causal to the phenotype.

In this study, we first performed a comprehensive analysis of the most recent meta GWAS of 20,406 UF cases and 223,918 controls[12]. We integrated the findings with epigenomic maps from the Encyclopedia of DNA Elements (ENCODE)[20] database and the Roadmap Epigenomics Project[21], as well as physical 3D chromatin organization[22] and gene expression data from the Genotype-Tissue Expression (GTEx) database[23] to identify potential gene targets of the UF risk loci. To account for ancestry and population genetic diversity, we also included GWAS data from more diverse populations and comparatively analyzed with the GWAS data from ref. 12, which is mainly from a white/European ancestry population. We integrated the genetic risk loci with bulk and single-cell gene expression and serum protein levels as quantitative traits to reveal potential molecular traits downstream of genetic risk loci. Furthermore, for select risk loci, we performed locus-specific CRISPR-based epigenetic editing to identify functional gene targets of these loci and their epigenetic states. Lastly, we integrated these analyses with single-cell gene expression data[24] in normal and UF tissue, highlighting potentially causal cell types where the risk-loci associated genes are most associated.

## Results

### The integrative multi-omics analysis identifies novel risk loci and gene targets

To identify disease-associated, potentially causal genetic variants of UF, we started with a meta-analysis of the largest GWAS reported by ref. 12. The original association analysis of 8,662,096 SNPs identified 1172 genome-wide significant SNPs ($p < 5 \times 10^{-8}$), 29 lead SNPs, and 127 target gene candidates that are associated with UF. To identify target genes that may be directly affected by UF-associated variants, we performed an integrative analysis of this GWAS data with several multi-omics data sets, including epigenomic maps from ENCODE[20] and Roadmap Epigenomics Project[21], 3D chromatin organization map from Hi-C experiments[22], and GTEx gene expression data[23]. To perform this integrative analysis, we used FUMA, a data analysis platform that can integrate multi-omics data with GWAS hits to annotate, prioritize, visualize, and interpret GWAS results[25]. Critically, this multi-omics analysis identified 34 lead SNPs and 172 independent significant SNPs out of 1653 candidate SNPs (1172 with a $p < 5 \times 10^{-8}$ and 481 within high LD) that encompass 24 genomic regions as candidate risk loci for UF pathogenesis (Fig. 1a, Supplementary Data 1). Among the 34 lead SNPs (Fig. 1b), only two were in exonic regions: rs10929757 (*GREB1*) and rs16991615 (*MCM8*). Among the others, we observed eight intergenic variants, 18 intronic variants, two ncRNA exonic variants, two ncRNA intronic variants, one promoter variant, and one three-prime untranslated region (3′ UTR) variant (Supplementary Data 1). In addition to the 18 most proximal candidate genes (Fig. 1c), the FUMA analysis identified 394 genes as potential targets of the 34 lead SNPs (Fig. 1d, Supplementary Data 1). These genes are linked to the GWAS risk loci due to (i) genomic linear proximity (<10 kb), (ii) 3D genomic organization (Hi-C contact frequency), or (iii) expression quantitative trait loci (eQTL). Around 60% of the 127 GWAS target genes identified in previous UF GWAS were included in the FUMA-identified candidate gene list (Fig. 1d, Supplementary data 1). Notably, a significant portion of the de novo candidates that we identified were due to the inclusion of gene expression (*n* = 116 eQTL specific candidate genes) and 3D chromatin interaction data (*n* = 147 3D genome-specific candidate genes) (Fig. 1e), indicating the power of integrating multiple layers of omics data to identify targets of disease-associated genetic variants.

### Integrating GWAS-identified risk loci with gene expression and chromatin state data from UF tissue.

The 24 UF risk loci are heterogeneous in size, number of SNPs in high LD, and the number of putative target genes (Fig. 2a). The narrowest locus (lead SNP = rs117245733) consists of a single nucleotide, whereas the widest locus (lead SNP: rs149934734) is ~500 kb. Furthermore, the number of significantly associated SNPs and the number of genes within or associated with each locus is not directly correlated with the size of the risk locus. For example, the risk locus containing the lead SNP rs78378222 is a relatively medium-sized genomic region (~300 kb), but it has one of the fewest numbers of significant SNPs. Interestingly, however, this locus contains the highest number of associated genes through eQTL, genomic position, and 3D chromatin interaction data.

To assess whether the identified risk loci are aberrantly regulated in a UF, we overlaid recently published (GSE128242) H3K27Ac chromatin state data (ChIP-Seq) and 3D chromatin interaction data (Hi-C) from UFs and matched normal myometrium of fibroid tumor (MyoF)[26]. Histone H3 Lysine 27 Acetylation (H3K27ac) is a histone modification that marks active enhancers and promoters in the genome[27,28]. Notably, six risk loci had differential H3K27ac signal intensities at the associated gene promoters, but 16 of the risk loci had significantly higher chromatin contact frequency in fibroid tissue (Fig. 2b), indicating that many risk loci are aberrantly regulated in disease settings at the chromatin organization levels.

The ultimate goal of GWAS is to identify altered molecular mechanisms that can explain the disease phenotype. To this end, linking disease variants with gene expression changes revealed many expression quantitative trait loci (eQTL) genes. Additionally, assessing tissue-specific protein levels and linking them to the disease risk loci may further reveal direct molecular mechanisms of disease-associated genetic variants. While it is challenging to quantify proteins in all

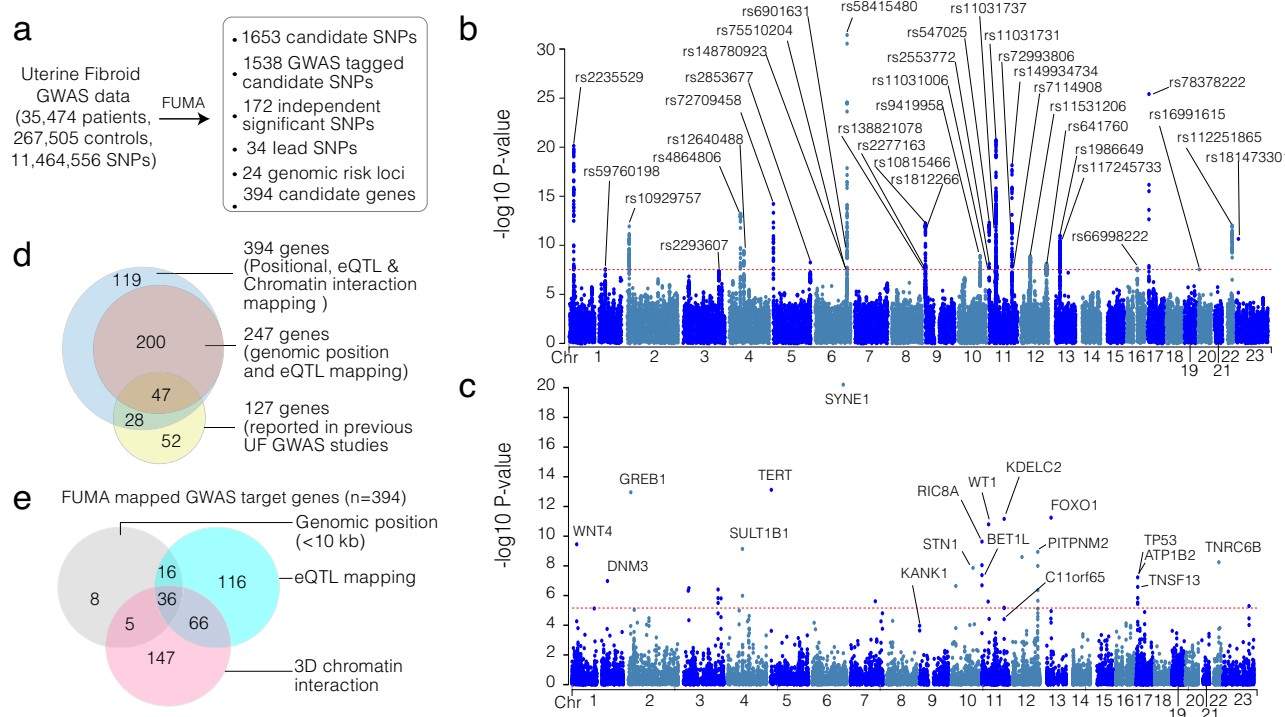

**Fig. 1 | Integrative analysis of uterine fibroid GWAS analysis with multi-omics data identifies novel candidate effector genes. a** Schematics show the strategy of integrating UF GWAS hits with genomic position, gene expression, and 3D chromatin interaction data using the FUMA data integration platform. The major findings are summarized in the box. **b** The Manhattan plot shows the significance (-log10 p value) between each SNP with UF disease phenotype. **c** The most likely gene targets of each genomic risk loci are shown. **d** The Venn diagram shows the number of candidates for UF GWAS target genes in this study and previous studies.

**e** The Venn diagram shows how the 394 genes in this study were mapped (i.e., linear genomic position, eQTL or 3D chromatin interaction) as a potential candidate target for the UF GWAS. For the Fig. 1b, c, the P values from summary statistics were originally calculated using Linkage Disequilibrium Score Regression (LDSR) and thresholded at a locus-wide significance of P < 1e10^-5 (Bonferroni correction). Gene annotation P values are shown with a P < 2.524e-6 Bonferroni correction threshold marked in red.

tissues, quantitative measurements of serum proteins in large population cohorts revealed a significant number of proteins as quantitative trait loci (pQTL) whose levels show a significant linkage to the disease association variant[29,30]. Using a recent large pQTL dataset conducted in 35,559 individuals assessing 4907 proteins in serum[30], we identified 5 of our candidate UF lead SNPs within +−20 kb of a pQTL-associated variant (Fig. 2c). We reasoned that this proximity means that the lead SNPs and the pQTL variants are within the same risk loci. Collectively, we identified 24 pQTL-associated variants as proximal to the risk loci that we identified in the FUMA analysis. Interestingly, some of the lead variants, such as rs78378222, are associated with multiple pQTL-associated variants. Of the 7 pQTL found proximal to rs78378222, C1QTNF1 (complement C1Q tumor necrosis factor-related protein 1) was also found to be significantly downregulated in leiomyoma relative to myometrium in 2 independent UF RNA-seq datasets[26,31]. Notably, although the levels of these pQTLs were measured in the serum of normal people, several of these proteins may play a critical role in UF pathology. For example, INHBA, a member of the TGF-β superfamily, is highly expressed in UF and is associated with enhanced extracellular matrix deposition and poor outcome[32,33]. Additionally, oxidative stress-involved glutathione peroxidase 1(GPx1)[34], autophagy regulator ATG4 family members MAP1LC3A, MAP1LC3B and GABARAP[35], and human ribosomal protein RPS19[36] are all known to be differentially expressed and associated with UF pathology.

We next wanted to know whether the genes that our integrative analysis identified as UF GWAS risk loci target genes are differentially expressed in UF tumors. We analyzed differential gene expression in two publicly available RNA-seq datasets (GSE128242[26] and GSE169255[31]) obtained from 15 and 6 patient UF tumors and matched

patient myometrium (MyoF) (Fig. 2d). We found that a significant number of UF risk loci target genes were differentially expressed (168/ 394 risk loci target genes, Fisher exact test, p = 6.235e-16 with an odds ratio of 2.74 and p < 2.2e-16 with an odds ratio of 3.73, for each data set, respectively) (Fig. 2e). Of these, 87 were significantly upregulated (FDR < 0.05), and 81 of them were downregmethulated (Fig. 2f) in UF tumors. Notably, none of the DEGs had contradictory expression patterns across data sets.

Although the Gallagher GWAS dataset is from a large pool of individuals, these individuals are from a relatively homogeneous ancestral background of European descent. We, therefore, integrated two additional GWAS data sets; the UK Biobank (over 20,000 individuals across six continental ancestry groups) and the Japan Biobank (~260,000 individuals of mainly Japanese ancestry). It is important to note that of the 24 risk loci identified by FUMA in the Gallagher et al. study, six were shared with the Japan biobank study and four with the UK biobank data set (Fig. 2g). To our surprise, only one risk locus was found in all three datasets, indicating the significance of considering diverse ancestral backgrounds when evaluating GWAS results.

**Integrative analysis of single-cell gene expression data reveals potential target cell types in UF.** One of the challenges in GWAS is identifying the causal cell type(s) in the heterogenous tissue contributing to the disease pathogenesis. Single-cell gene expression tools provide unprecedented power in revealing cell type-specific gene expression programs in healthy and diseased tissues[37]. We queried a single-cell RNA-Seq atlas from 5 MyoF controls and 5 *MED12*-mutant fibroid tumor tissues[24] for the 168 FUMA-identified genes that were differentially expressed in UF tumors according to the bulk RNA-Seq

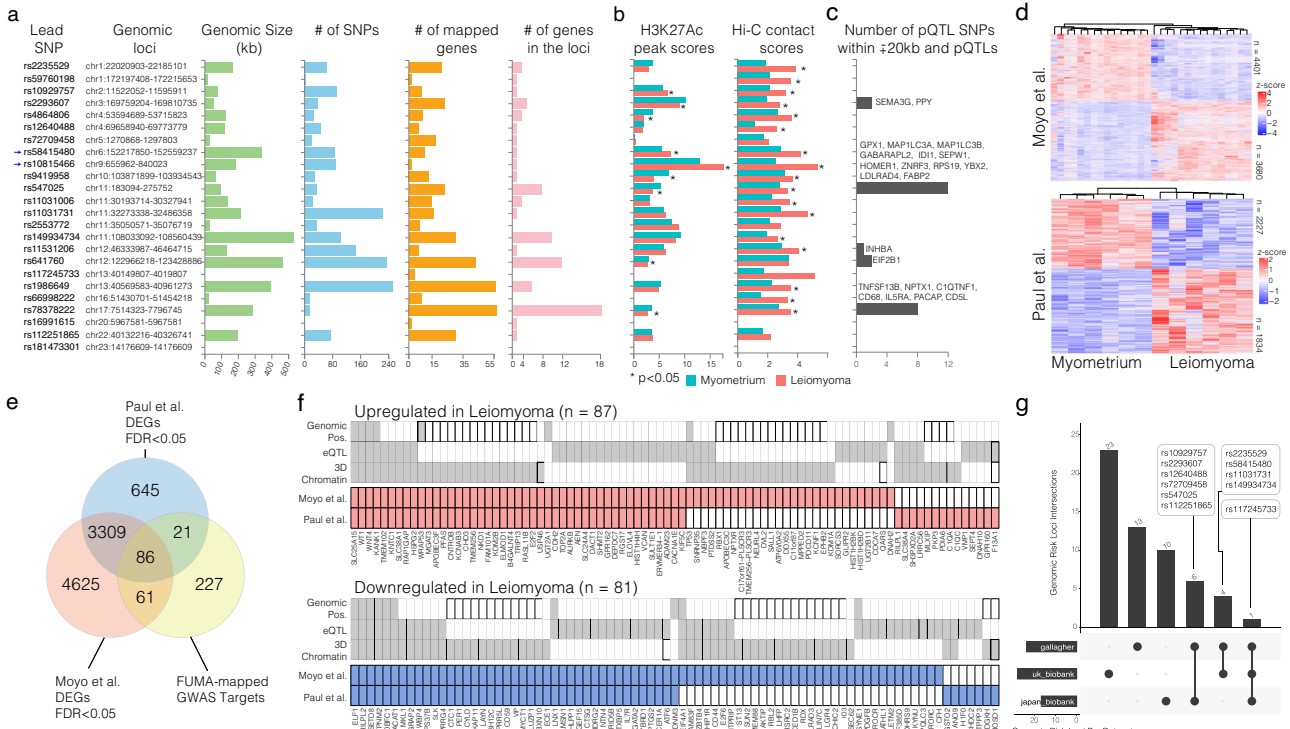

**Fig. 2 | Uterine fibroid GWAS risk loci are variable in size and the number of target genes that are differentially expressed in uterine fibroid tumors.**
**a** Horizontal bar graphs show genomic loci, size, number of SNPs, number of mapped genes, and number of genes in each of the 24 UF GWAS risk loci.
**b** Horizontal bar graphs show the relative intensity of H3K27ac ChIP-Seq signal intensity and Hi-C chromatin contact frequency in primary uterine fibroid tumors and matched MyoF tissue samples. **c** The horizontal bar plots indicate the number of pQTL-associated variants that are proximal to each locus. The names of each pQTL protein are indicated. **d** The heatmaps show the number of differentially

regulated genes between leiomyoma and MyoF in two independent bulk RNA-Seq datasets. **e** The Venn diagram shows the number of FUMA-identified target genes that are common between the differentially expressed genes from the indicated independent data sets. **f** The diagrams show the FUMA-identified genes that are upregulated (n = 87) or downregulated (n = 81) in leiomyoma in at least one dataset. **g** The upset plot shows the common set of UF risk loci across three different GWAS studies. The lead SNP from Gallagher et al. is indicated for the common risk loci. For the Fig. 2b, two-sided student t test was performed.

datasets. Our re-analysis of the scRNA-Seq atlas of MyoF and Leiomyoma tissues by ref. 24 revealed 6394 cells divided into 18 distinct cell clusters in MyoF and 41,864 cells divided into 17 clusters in UF tumor tissue (see methods for the QC steps)[24]. These cell types can be broadly grouped into smooth muscle cells (SMC), fibroblast, endothelial, lymphoid, and myeloid cell clusters (Fig. 3a). These cell types were identified using the same marker genes used by Goad et al.: *MYH11* for SMCs, *DCN* for fibroblasts, *PECAM1* for endothelial cells, *CD3D* for lymphoid cells, and *CD14* for myeloid cells, as well as GO term analyses using the top enriched markers for every cluster (Supplementary Figs. 1 and 2). The single-cell data also allows for examining how the overall tissue architecture and cellular composition change between a normal and disease state. Critically, in line with known disease pathology[1,38], we noted a drastic increase in the percentage of SMC cells in UF tumors (from ~14% in myometrium to ~65% in Fibroids) (Fig. 3b).

We next aimed to identify potentially causal cell types where GWAS-identified targets are most highly expressed. Significantly, the SMC, the cell type of origin for UF tumors, contain the highest number of highly expressed GWAS target genes in the leiomyoma samples. Around 40% of GWAS-targets genes that are differentially expressed in leiomyoma tumors (33/82) are highly expressed selectively in the SMC cluster (Fig. 3c). Other clusters contained a substantially smaller number of highly expressed genes, but each has its own module of enriched candidate genes. For example, the *KANK1* gene, a UF risk locus target gene upregulated in UF tumors[12] and involved in actin polymerization and cytoskeleton organization, is explicitly expressed predominantly in SMC (Fig. 3d). Conversely, *VCAN*, an upregulated

gene in UF tissue[26] and codes for sulfate proteoglycan, a major component of the extracellular matrix, is expressed specifically in fibroblast cells, which is another critical cell type that cooperates with mutant SMC to drive leiomyoma pathogenesis[39] (Fig. 3d). We have also noted that a set of GWAS targets is specifically expressed in other cell types. For example, the *C1QC and C1QA* genes, which code for the complement C1Q C and A chains, are highly expressed predominantly in lymphoid cells in the leiomyoma tissue, indicating a potential link between the immune system and UF pathogenesis[40] (Fig. 3d). Importantly, many of these genes are detected in a larger proportion of the respective cell types, stressing their cell type specificity (Supplementary Fig. 3).

**Locus-specific epigenetic editing to map risk loci target genes.** The integrative multi-omics analysis substantially expanded the number of putative risk loci target genes. It is plausible that some of these target genes may be false positives. To experimentally validate the target genes, we utilized the CRSPR-based locus-specific epigenetic editing tools to functionally link GWAS risk loci with the target genes. By utilizing the catalytically inactive CRISPR/Cas9 (dCas9)[41,42], we aimed to recruit the repressive KRAB domain that leads to the accumulation of repressive Histone H3 Lysine 9 methylation and gene repression at the target locus. Therefore, we designed multiple lead SNP targeting sgRNAs and control sgRNAs that target genomic regions distal to any of the GWAS SNPs and that are not regulatory elements (i.e., no H3K27ac mark). We utilized the hTERT-immortalized myometrium SMC line[43], a UF-relevant cell type that retains certain aspects of human myometrial function[44]. To identify the relevance of these loci under

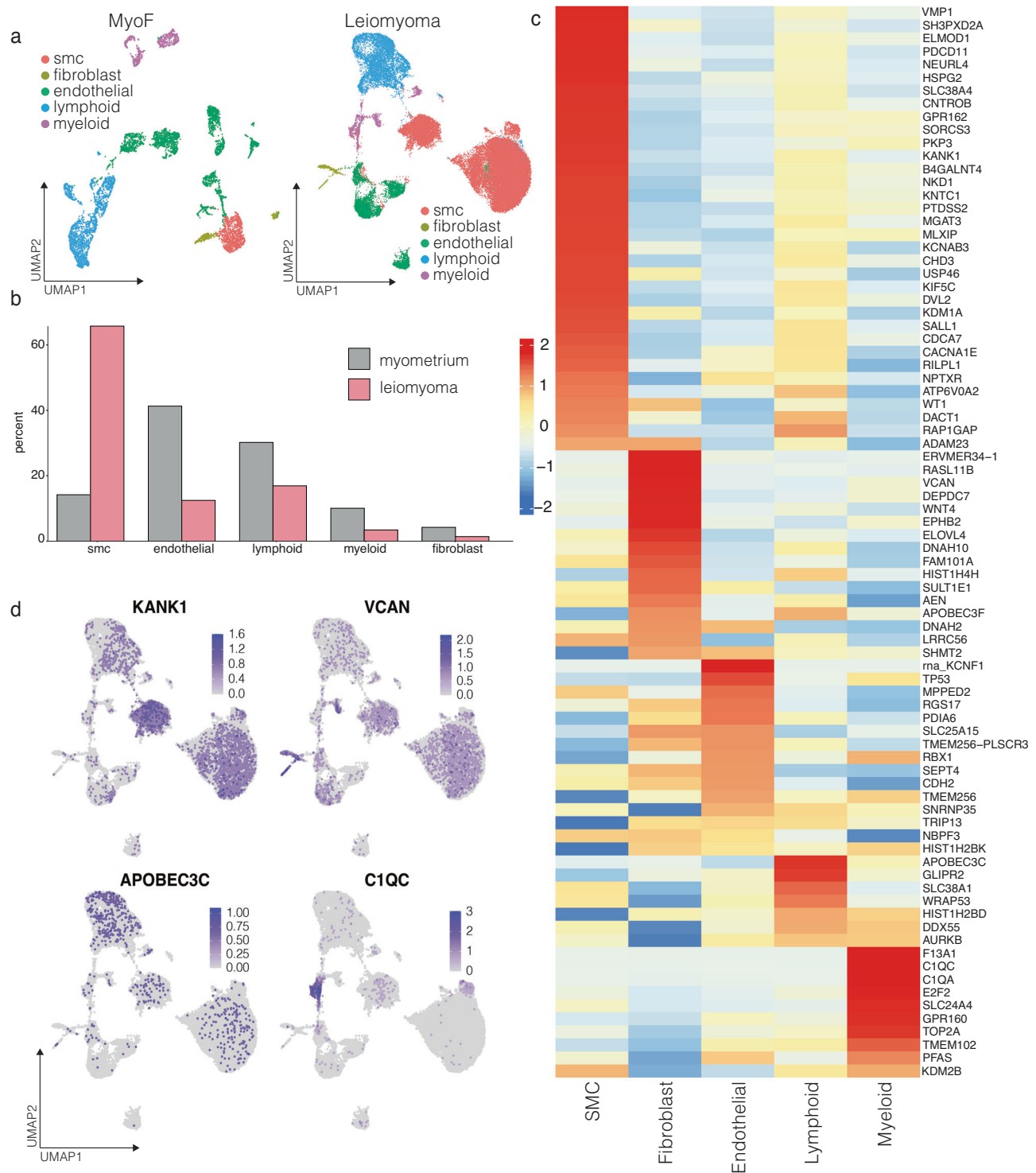

**Fig. 3 | Integrating GWAS risk loci with single-cell expression profiles in relevant normal and disease tissue reveals potential cell type targets of risk loci genes. a** The dot plot represents uniform manifold approximation and projection (UMAP) of single-cell gene expression (scRNA-Seq) data in MyoF and uterine fibroid tissue. Relevant cell types are labeled based on select marker genes that were identified through differential expression analysis[24]. **b** The bar plots show the percent of MyoF and leiomyoma tissue represented by indicated cell types. **c** The heat map shows cell type-specific expression levels of differentially expressed UF GWAS target genes in single-cell data of MyoF (**b**) or Uterine Fibroid tissue (**c**). The genes are separated into upregulated (**red**), or downregulated (**blue**) genes based on the differential expression in bulk RNA-Seq data from Uterine fibroids compared to MyoF in Fig. 2f. The color of each is determined by row z-scores. **d** The heat in the dot plots (feature plots) shows the relative normalized expression of the indicated genes in various cell clusters from the UMAPs of Uterine fibroid tissue.

investigation, we also performed H3K27ac ChIP-Seq analysis in these cells to map potential active enhancers and promoters in this cell type.

We initially targeted the rs58415480 lead SNP locus (Fig. 4a), a well-established UF risk locus consistently identified in multiple GWAS studies in UF, breast cancer[45], and osteoporosis phenotype[46,47].

Notably, the lead SNP and multiple other SNPs in high LD ($r^2 > 0.8$) are located in a regulatory element in the *SYNE1* gene body. This likely enhancer element is marked with the H3K27ac, which is further enhanced in UF fibroids compared to the myometrium, indicating that this regulatory element is further activated in UF. *SYNE1* encodes a

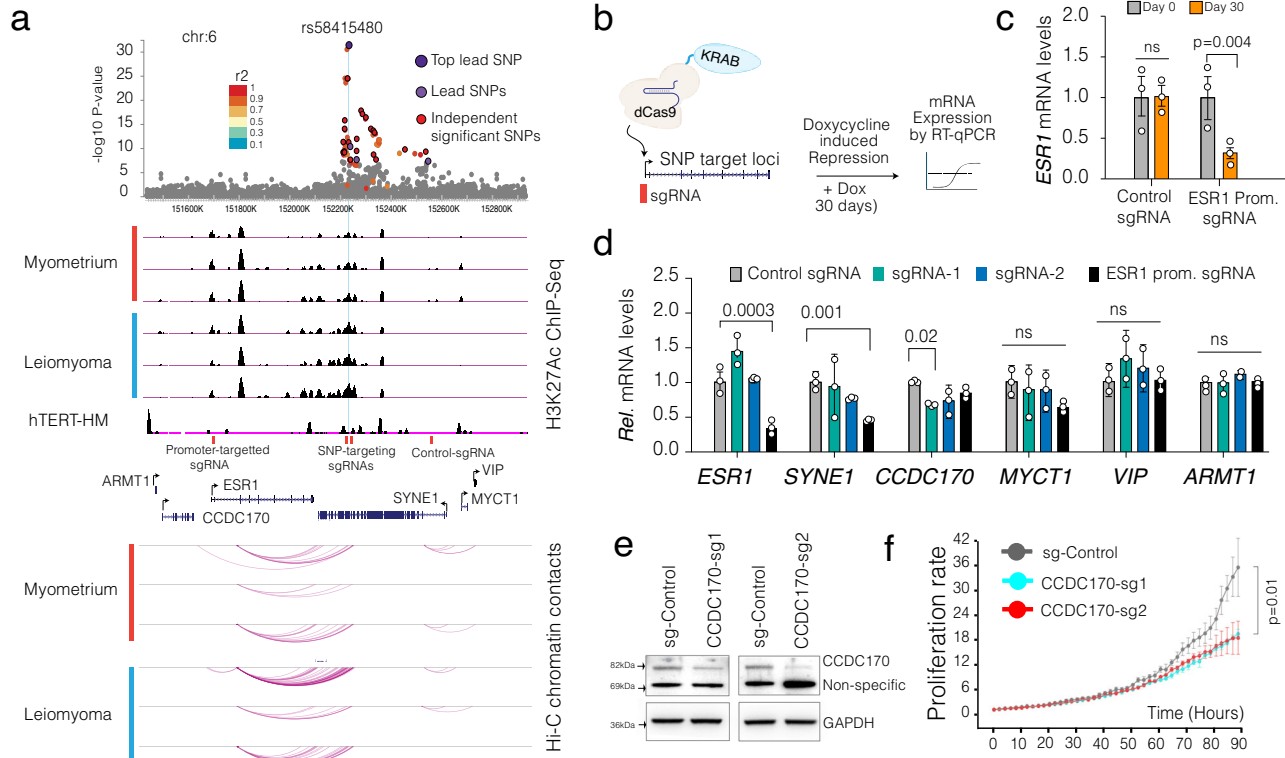

**Fig. 4 | Locus-specific epigenetic editing to identify lead SNP region target genes. a** The dot plot shows the significance of SNP in the rs584154480 lead SNP risk loci. ChIP-Seq track shows the signal intensity of H3K27ac and arc plots show the intensity of Hi-C measured chromatin contact frequency in the same genomic region for UF tissue and MyoF patient samples. **b** Schematics show locus-specific recruitment of repressive dCas9-KRAB protein strategy and assessment of gene expression changes. **c** The bar plot shows ESR1 mRNA levels in dCas9-KRAB expressing smooth muscle cells with control sgRNA and ESR1 promoter targeting sgRNA. **d** The plots show mRNA levels of indicated genes in dCas9-KRAB-

expressing smooth muscle cells with indicated sgRNAs. **e** Western Blot shows protein levels of CCDC170 in cells expressing control and CCDC170 targeting sgRNAs. Uncropped raw membrane is shown in the Supplementary Fig. 6. **f** The plot shows relative cell proliferation rate in cells expressing control sgRNA and CCDC170 targeting sgRNA as measured by Incucyte live-cell imaging platform. Two-tailed student *t* tests were used for all statistical comparisons. Error bars in Fig. 4c, d, f indicate the standard error of the mean of three independent biological replicates (*n* = 3).

---

spectrin repeat-containing protein that localizes to the nuclear membrane and is highly expressed in skeletal and SMCs. Mutations in this gene have been associated with spinocerebellar ataxia and bipolar disorder[48,49]. Whether or not the rs58415480 risk locus contributes to UF pathogenesis due to altered expression of *SYNE1* is currently unknown. Notably, the plausible causal gene candidate in this risk locus is *ESR1*, the most proximal gene to *SYNE1*. *ESR1* encodes for estrogen receptor, a ligand-activated transcription factor. Estrogen hormone and its receptor are known regulators of UF pathogenesis[3,50–52], and *ESR1* is significantly upregulated in UF in both of the RNA-seq datasets that we analyzed. Significantly, Hi-C chromatin interaction maps in myometrium show substantial 3D genomic interaction between the lead SNP-containing genomic region in the *SYNE1* gene and *ESR1* gene (Fig. 4a), highlighting the interaction between these proximal genes. We, therefore, initially assessed the efficiency of our targeted epigenetic editing on the *ESR1* gene itself by using doxycycline (dox)-inducible dCas9-KRAB (Fig. 4b). After recruiting dCas9-KRAB to the *ESR1* promoter with a sgRNA, we observed a significant repression of *ESR1* after two weeks of Dox treatment and dCas9-KRAB expression (Fig. 4c). As a negative control, we utilized a control sgRNA that targets a non-regulatory region ~80 kb away from the *ESR1* promoter. Significantly, among other genes tested, we only observed significant repression of *SYNE1*, indicating potential interaction between these two promoters. which are more than 900 kb away from each other (Fig. 4d).

We next designed two sgRNAs targeting rs58415480 and measured the expression of all six genes in this particular risk loci. Notably,

the only gene that significantly downregulated when the lead SNP region was targeted was *CCDC170*, which encodes for a coiled-coil domain-containing protein. These results indicate that *CCDC170* might be a gene directly linked to the GWAS target loci. Although little is known about the role of *CCDC170* in UF, it is recurrently fused with the *ESR1* gene in ~14 % of ER+ breast cancers, and patients with such fusion have worse clinical outcomes[53]. Studies in breast cancer cell lines have implicated *CCDC170* in cell migration through the alteration of the Golgi-associated microtubule network[54]. Given the lack of evidence for the role of *CCDC170* in fibroid pathogenesis, we used CRISPR to knock out this gene and study its phenotype (Fig. 4e). Using the Incucyte live cell imaging platform, we noted that *CCDC170* depleted cells (using multiple sgRNAs) proliferated significantly less compared to control sgRNA-expressing cells (Fig. 4f). Notably, we observed a comparable rate of cell death (Supplementary Fig. 4), indicating that *CCDC170* is a critical regulator of cell proliferation in myometrium SMCs. To our surprise, *ESR1* expression did not change when the lead SNP region was targeted, indicating that, at least in the hTERT-SMC we used, ESR1 is not directly regulated by the lead SNP-containing regulatory element.

We next targeted lead SNP, rs10815466, which resides in a regulatory element within the *KANK1* gene and is further activated in UF, as evidenced by higher H3K27ac levels and more frequent Hi-C contact frequencies (Fig. 5a). Our H3K27ac ChIP-seq data in the hTERT-SMC cells shows that this regulatory element is also active in this cell line. We designed two independent lead SNP targeting sgRNAs and a control sgRNA to epigenetically silence this regulatory element by recruiting dCas9-KRAB to these regions. We initially checked the level

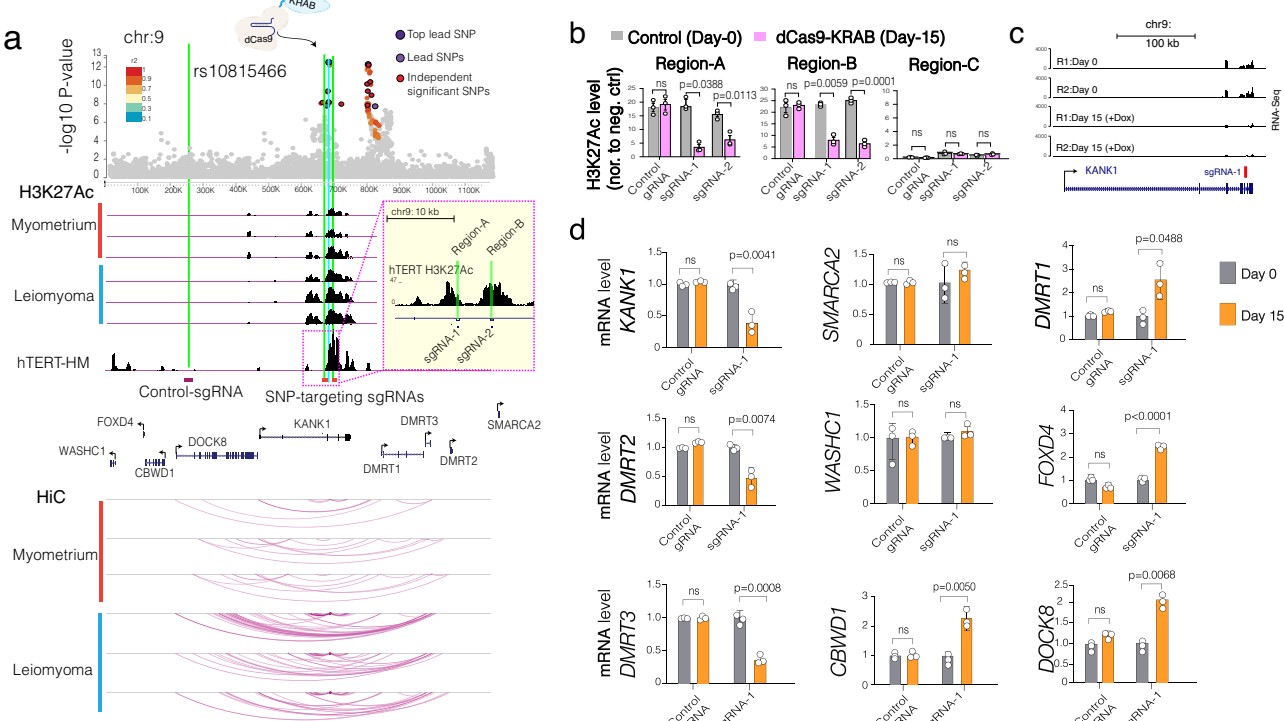

**Fig. 5 | Locus-specific epigenetic editing to fine map effector genes in rs10815466 risk locus. a** The dot plot shows the significance of SNPs in the rs584154480 lead SNP risk locus. ChIP-Seq tracks show the signal intensity of H3K27ac and arc plots show the intensity of Hi-C measured chromatin contact frequency in the same genomic region for UF tissue and MyoF patient samples. **b** The Bar plot shows ChIP-qPCR measured H3K27ac levels at the indicated region on Day 0 and 15 days after the induction of dCas9-KRAB. **c** RNA-Seq bigwig tracks show relative expression of KANK1 locus before and after 15 days of dCas9-KRAB

recruitment to the KANK1 target locus with sgRNA-1. Tracks heights are normalized to read depth. **d** The bar plot shows mRNA levels of indicated genes in dCas9-KRAB expressing smooth muscle cells with control sgRNA and sgRNA targeting around the rs10815466 lead SNP. The relative mRNA levels of all nine genes proximal or 3D linked to the rs10815466 lead SNP risk loci were measured by RT-qPCR. Two-tailed student t-tests were used for all statistical comparisons. Error bars in Fig. 5b, d indicate the standard error of the mean of three independent biological replicates (*n* = 3).

of epigenetic repression by assessing the H3K27ac levels at the control and the SNP targeting sgRNA regions (regions A and B in Fig. 5a). Chromatin immunoprecipitation and qPCR (ChIP-qPCR) analyses showed significant enrichment of H3K27ac levels at both regions A and B compared to the control region at the basal level. More importantly, after two weeks of doxycycline-inducible dCas9-KRAB expression, we observed a significant reduction of H3K27ac levels at both positive control regions exclusively with the lead-SNP-targeting sgRNA but not with the control sgRNA. This indicates that our locus-specific epigenetic repression strategy works specifically and robustly (Fig. 5b). We then performed RNA-Seq (with sgRNA1, which led to more robust repression) to ensure that the target region is differentially expressed. As shown in (Fig. 5c), we observed substantial repression of KANK1 gene based on normalized RNA-Seq bigwig tracks.

Encouraged by these results, we assessed gene expression changes in all other genes proximal to the KANK1 risk loci using more sensitive RT-qPCR as several neighboring genes were below detection based on the depth of RNA-Seq (~15 million reads). Notably, in addition to KANK1 itself, we observed significant repression of DMRT2 and DMRT3, which are 470 kb and 585 kb away from the target site, respectively (Fig. 5d). Conversely, we did not see any change in the expression of the two most distal genes (SMARCA2 and WASHC1). To our surprise, we also observed significant upregulation of genes that are the most proximal to the lead SNP target site, including DMRT1, FOXD4, CBWD1, and DOCK8. These findings highlight the complexity of this regulatory element and indicate that the activation of this lead-SNP-containing regulatory element is positively regulating the expression of KANK1, DMRT2, and DMRT3 while simultaneously negatively regulating the activity of 4 other genes that are more proximal to

the regulatory element containing lead SNP. Comparable results were obtained using the second sgRNA targeting the risk loci (Supplementary Fig. 5).

The above findings support the utility of locus-specific epigenetic repression to study potential gene targets. However, such an approach may not be helpful if the lead SNPs containing regulatory element is inactive in the cell type of choice. In such cases, we hypothesized that locus-specific epigenetic activation could be used to map the potential gene targets. To demonstrate a proof of principle, we used a dCas9-based strategy to recruit P300 histone acetyltransferase (dCas9-P300)[55,56] to specifically deposit the H3 lysine 27 (H3K27ac) histone modification, an epigenetic mark associated with active enhancers and promoters[27,28], to an inactive locus. We chose to target the lead SNP rs78378222, located at the end of the TP53 5'-UTR region with some enhancer activity based on H3K27ac levels in patient samples but is inactive in the hTERT-SMC cells we used (Fig. 6a). The 3D Hi-C contact frequencies indicate that the risk locus contains 3D chromatin contacts, specifically in leiomyoma patient samples compared to MyoF (Fig. 6a). Several of these 3D contacts are between the TP53 promoter region and the distal genes, such as TNFSF12, EIF4A1, and ATP1B2 genes, which are the genes that were identified by FUMA as putative targets of this risk loci. We recruited dCas9-P300 with two independent sgRNAs targeting the SNP region and assessed the enrichment of the local H3K27ac levels (ChIP-qPCR) compared to a control sgRNA, targeting Luciferase gene. As expected, the recruitment of P300 resulted in more than 15-fold enrichment of H3K27ac levels at the SNP target site (Fig. 6b). After observing this robust activation, we also performed RNA-Seq and observed substantial upregulation of the target locus when dCas9-P300 was recruited with one of the sgRNAs

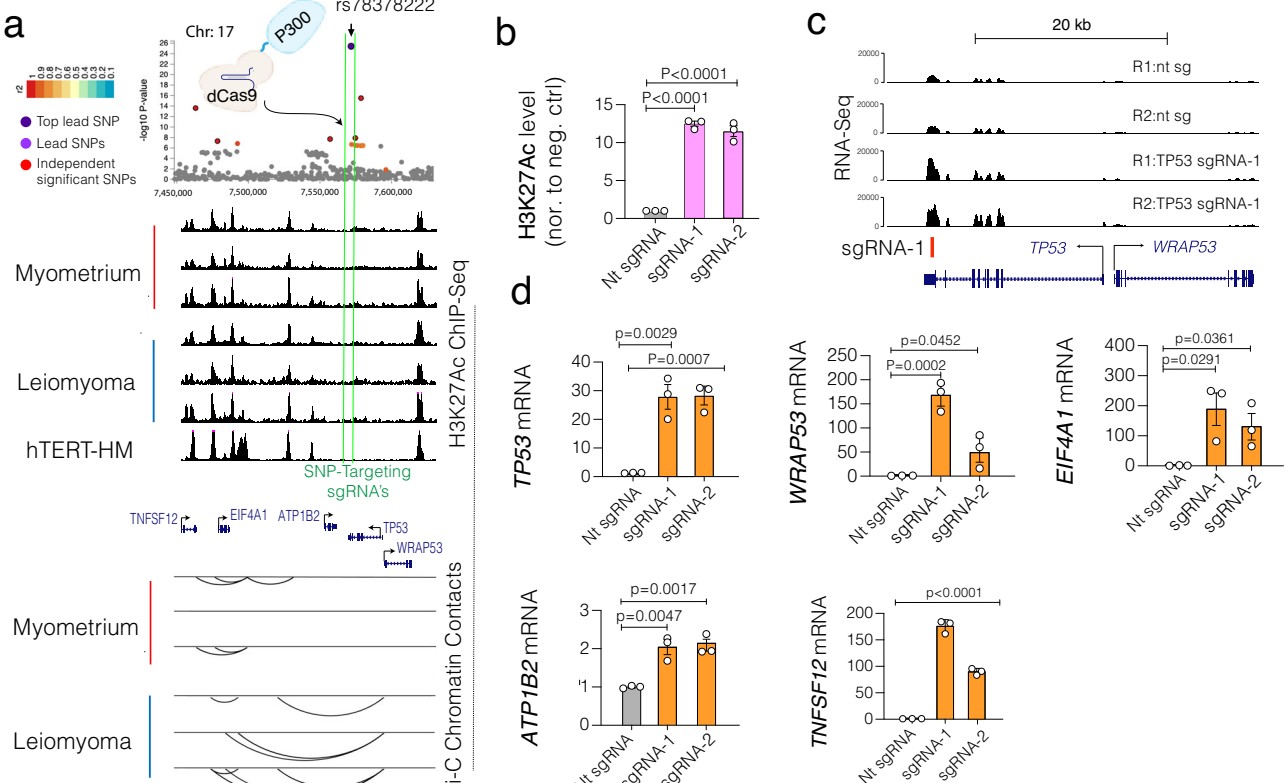

**Fig. 6 | Locus-specific epigenetic editing to upregulate rs78378222 risk locus to validate the target genes linked to the risk locus. a** The dot plot shows the significance of SNPs in the rs78378222 lead SNP risk locus. ChIP-Seq track shows the signal intensity of H3K27ac and arc plots show the intensity of Hi-C measured chromatin contact frequency in the same genomic region for UF tissue and MyoF patient samples. Track heights are normalized to read depth. **b** The bar plot shows qPCR measured enrichment of H3K27ac chromatin immunoprecipitation levels at the target locus after recruitment of dCas9-fused P300 histone acetyltransferase with control sgRNA (sgRNA-targeting another chromosome) and lead SNP

targeting sgRNA. **c** RNA-Seq bigwig tracks show relative expression at *TP53* locus after recruitment of dCas9-P300 epigenetic activator with control and TP53 locus targeting sgRNA-1. Tracks heights are normalized to read depth. **d** Bar plots show RT-qPCR measured mRNA levels of indicated genes with and without recruitment of dCas9-P300 to the lead SNP target region. Two-tailed student t-tests were used for statistical comparisons. Error bars indicate the standard error of the mean. Error bars in Fig. 6b, d indicate the standard error of the mean of three independent biological replicates ($n = 3$).

(Fig. 6c). Encouraged by these findings, we then measured the gene expression changes in the five FUMA-identified putative target genes, including the *TNFSF12* gene, located ~120 kb away from the target site, using a more sensitive RT-qPCR assay. Critically, all five genes show variable and significant upregulation upon recruitment of dCas9-P300 (Fig. 6d), validating that the transcriptional activity of these genes is regulated distally by the epigenetic state of the GWAS risk loci.

## Discussion

UFs affect ~70% of women. In 15–30 % of these patients, these tumors interfere with normal uterine functions and cause a significant emotional and financial burden on the individual and society at large. Notably, relatively few recurrent somatic mutations have been observed in UF tumors compared with other cancers. Nearly 70% of all UF tumors are driven by recurrent somatic mutations in the *MED12* gene, encoding the MED12[4]. Other recurrent somatic alterations are translocations in the *HMGA 1&2* genes, loss of *FH*, deletion of the *COL4A5-COL4A6* gene and mutations of SRCAP complex that leads to defective deposition of H2AZ histone variant[5,6]. Critically, in addition to these recurrent somatic mutations in select genes, heritable genetic variations are believed to contribute to UF pathogenesis significantly. Several GWAS have identified specific germline genetic variants and genomic loci significantly associated with UF[12–17]. How and whether the recurrent somatic mutations cooperate with the existing germline variants are currently unknown and should be an area of active investigation in the future.

In this study, we utilized the FUMA data integration platform[25] to re-analyze the largest UF-specific GWAS data by ref. 12. Our findings have significantly expanded the number of genes potentially regulated by the germline genetic variations called risk loci. Previous GWAS had implicated 127 genes as UF-specific GWAS risk loci targets. In contrast, our integrative analysis of multi-omics data, including 3D chromatin interaction data, has greatly expanded the number of putative gene targets to 394. Even though this is likely an overestimation, the fact that more than 40% of these genes are differentially expressed between MyoF and fibroid tumor tissue indicates that a substantial number of these genes could be the direct target of risk loci and contribute to the disease pathology.

Our study aimed to address several challenges in GWAS. First, a vast majority of the GWAS-identified lead SNPs and independently significant SNPs are not located within protein-coding regions of genes but rather in the non-coding regulatory elements. Second, GWAS risk loci can span multi-megabase genomic regions, creating a formidable challenge to directly link GWAS loci to a specific gene target and fine-map the causal gene targets of the genetic variation. The third challenge is to infer the cell types within which the GWAS-identified risk loci target genes manifest and may cause disease progression[57–59].

We integrated multi-omics data with the largest UF-associated GWAS analysis to address the first and second challenges. This has significantly expanded the gene targets associated with the risk loci. Although the integrative analysis significantly expanded the number

of target genes, figuring out which of these genes are directly regulated by the risk loci and are causal to the disease phenotype remained a significant challenge. In order to narrow down this list of genes, we utilized locus-specific epigenetic editing to manipulate the epigenetic state of the target loci and measure the gene expression changes of any target genes associated with the loci. Significantly, this approach revealed complex regulatory networks at three distinct GWAS loci in fibroid tumors. For example, we found that the epigenetic manipulation of the *ESR1* gene promoter directly affects the expression of the *SYNE1* gene, which is nearly 900 kb away from the *ESR1* promoter. Notably, when the SYNE1 risk locus was epigenetically perturbed, ESR1 did not change, but we observed a significant change in the expression of *CCDC170*, indicating, at least in this cell line, *that CCDC170* is a critical risk loci target gene, although it is not an eQTL gene. Notably, recent detailed analysis indicates that eQTL loci explain only a small fraction of GWAS loci because these two variants are systematically different from each other[60], highlighting the limitation of relying exclusively on eQTLs to explain GWAS risk loci targets and the significance of functional perturbations to map GWAS targets genes.

To overcome the third challenge, we exploited the single-cell gene expression analysis of MyoF and UF tumors to identify potential cell types impacted by the differential expression of GWAS target genes. Despite the sparsity of gene expression data in scRNA-seq, our integrative analysis revealed several interesting observations. We found that the SMC, the causal cell of origin for the disease progression, show the highest number of highly expressed putative GWAS risk loci target genes. Furthermore, the data indicates that almost all cell types composing the UF tumor tissue contain distinct modules of genes highly expressed in a cell type-dependent manner. For example, while *KANK1* is substantially upregulated predominantly in SMCs, *VCAN* is highly expressed selectively in fibroblasts in fibroid tumors. Whether and how differential expression of these genes is associated with fibroid tumors is yet to be understood.

Our study has some limitations. Firstly, we have focused most of our efforts on studying the risk loci identified in one GWAS study. We find that the GWAS-identified risk loci are highly dependent on the genetic ancestry of the population under study. Therefore, further studies are needed to comprehensively integrate and study the gene targets of multiple GWAS data that represent diverse populations. Secondly, although our analysis substantially expanded the number of target genes, confirming that these genes are directly linked to the GWAS-risk loci requires experimental validation including precise genome editing of the alternative alleles. Such experimental systems should ideally capture the complexity of in vivo tissue. However, most in vitro systems fall short of this ideal. We perturbed the epigenetic state of the three GWAS risk loci in a SMC line, which may not represent the complex regulatory network of a GWAS loci in vivo. However, if the regulatory elements (promoters and enhancers) in a particular locus have high activity in the tissue of interest, the dCas9-KRAB approach could be used to downregulate the target locus and study gene targets. Conversely, if the lead SNP target site is inactive in the model cell line, then dCas9-P300-like epigenetic activation approaches could be used to map potential target genes linked to the target site's epigenetic state. In an ideal world, such perturbations should be performed in a model system representing the disease state. However, there is a lack of tractable model systems that recapitulate UF disease phenotypes. To address this, we recently utilized CRISPR to engineer UF-specific genetic mutations in the MED12 gene. These engineered cells represent a new in vitro model for UF research by recapitulating critical transcriptional and metabolic programs of UF tumors[38]. High throughput single-cell genetic and epigenetic manipulations such as Perturb-Seq[61,62] in such model systems should allow us to better map and validate GWAS risk loci targets and study their relevance to disease biology.

## Methods

### Uterine Leiomyoma GWAS summary statistics

Summary data of UF GWAS were downloaded from the NHGRI-EBI GWAS Catalog[63] for study GCST009158[12] downloaded on 06/05/2020. 11,464,556 SNPs were used for post-UL GWAS analyses and uploaded to FUMA (version 1.3.6)[25]. Round 2 UK Biobank GWAS summary statistics were downloaded from the Neale lab/UK Biobank website: http://www.nealelab.is/uk-biobank on 05/18/2023. Japan Biobank GWAS summary statistics were downloaded from the Japan Biobank website: https://biobankjp.org/en/ https://pheweb.jp/ on 06/12/23. Both UK Biobank and Japan Biobank GWAS summary statistics were uploaded to FUMA (version 1.3.6) and analyzed using the same settings as the Gallagher et al. data (see below).

### Identification of candidate genes using FUMA

Independent significant SNPs are defined as all SNPs in GWAS summary data that have $P$ value < 5e-8 and are independent each other at $r^2 < 0.8$. Among independent significant SNPs, Lead SNPs were defined as SNPs that are distanced from each other at $r^2 < 0.1$. Independent significant SNPs that are dependent upon one another at $r2 \geq 0.1$ participate in the same genomic risk loci and LD blocks closer than 250 kb merge into same genomic risk loci. Candidate SNPs are all the SNPs that are in the LD of any independent significant SNPs ($r2 > 0.8$). To calculate $r^2$, MAF (minor allele frequency) and conduct LD analyses, European population genetic data in 1000 G phase3[64] were selected for reference panel. Also, MAGMA gene and gene-set analyses were performed for the GWAS summary statistics. (MAGMA v1.07, 0 kb window)

Three methods were used to identify candidate genes (positional mapping, expression qualitative trait loci mapping and 3D chromatin interaction mapping) and every annotation reference dataset available on FUMA was used. First, for positional mapping, candidate genes were identified that are within 10 kb of candidate SNPs. Then, for eQTL mapping, the genes are selected based on if SNPs in genomic risk loci significantly affect (FDR < 0.05) the gene's expression in all eQTL databases available in FUMA. 3D chromatin interaction mapping was performed using all HI-C datasets in FUMA and significant interactions were chosen using a FDR < 1e-6. Interactions were chosen only between SNPs which overlap with enhancer regions and genes promoter regions as defined by the default parameters of FUMA (250 bp upstream and 500 bp downstream of TSS). All FUMA-available epigenome datasets were selected to annotate enhancer/promoter regions. FUMA uses eQTL datasets of more than 44 tissue types and uses HI-C data sets of 21 tissue/cell types for genome interaction mapping. The original FUMA[25] publication is referred for details on all databases, depositories, and methods.

pQTL analysis: pQTL summary statistics were downloaded from https://www.decode.com/summarydata/. pQTL-associated variants that were within a 40 kb (+/−20 kb) window of each lead SNP in the Gallagher GWAS were identified in R.

### Cell lines

The human myometrial SMC line myo-hTERT Cells were kindly provided by Dr. Jian-Jun Wei (Northwestern University) and are described by Carney et al.[43]. The cell line is listed in the Cellosaurus database under the accession number CVCL_9Z20. The cells were maintained in DMEM/F-12 (Gibco, Invitrogen 11320033) with 10% Fetal Bovine Serum (Fisher scientific, SH3091003) and 1% Penicillin–streptomycin (Life Technologies,15140–122). Cells were cultured and incubated at 37 °C in a humidified atmosphere of 5% CO2 and 95% air.

### Locus-specific epigenetic editing by dCas9-KRAB or dCas9-P300

All sgRNAs were cloned into a modified sgRNA scaffold backbone from the GeCKO human library. BsmB1 digestion was used for cloning. And pHAGE TRE dCas9-KRAB (Addgene, Plasmid #50917) was used to generate doxycycline inducible dCAS9-KRAB human myometrial

hTERT cells. All sgRNA sequences and their oligos are listed in Supplementary data 2. Newly synthesized constructs were confirmed with Sanger sequencing before transfection.

For viral production, HEK293T cells were seeded onto 10 cm plates 1 day before transfection. 1 μg pMD2.G (Addgene, Plasmid #12259), 2 psPAX2 (Addgene, Plasmid #12260) and 4 μg of the guide plasmid were co-transfected into HEK293T cells using 21 μg of polyethylenimine (PEI). Media was refreshed 12 h after transfection. The virus was collected 24 and 48 h after the first media refreshment, filtered through 0.45 mm filter, and stored at −80 °C. For viral transduction, cells were incubated with virus solution diluted in media and supplemented with 0.01 mg/mL polybrene for 24 h. Human myometrial hTERT cells were transduced with lentivirus carrying the LV-dCAS9-KRAB plasmid and were selected with G418 (400 μg/mL) until non-transduced control cells were all dead (~5 days). Then, dCAS9-KRAB hTERT cells were transduced with lentivirus carrying the LV-sgRNA plasmid and were selected with Zeocin (100 μg/mL) until non-transduced control cells were all dead. After selection, cells were induced with doxycycline (2 μg/mL) to activate dCAS9-KRAB.

For dCas9 P300, the human myometrial hTERT cells were transduced with lentivirus carrying the LV-sgRNA plasmid targeting the lead SNP and Luciferase (control sgRNA) and selected with Zeocin (100 μg/mL) until non-transduced control cells were dead. Lastly, LV-sgRNA plasmid targeting the lead SNP and Luciferase (control sgRNA) cells were transduced with lentivirus carrying the LV-EF1a-dCas9-P300-PuroR (Addgene, Plasmid #83889) plasmid and selected with Puromycin (2.5 μg/mL) until non-transduced control cells were dead.

## Incucyte live cell imaging

Incucyte Live cell imaging system (Sartorius) was used for tracking cell proliferation. The system took a photo of cell plates every two hours in different image channels (Phase and Red). For cell nucleus counting, 1 μM SiR-DNA nuclear dye was used (Cytoskeleton, #SC007) and images were captured using the red channel. At the end of the experiment, proliferation data were analyzed using the Incucyte analysis tool and $p$ values were calculated using the Incucyte raw data. Relative proliferation was normalized to the starting time. For the Apoptosis assay, cells were seeded into 96-well plates at a density of $1.5 \times 10^3$ cells/well. The following day, 1:1000 diluted Caspase 3/7 dye (10403, Biotum) was added. Then, cells were monitored using the Incucyte live cell imaging system using phase and green channels. The apoptosis rate was determined using the green integrated intensity/confluency values, and the results were plotted using the Incucyte.

## Western blotting

Cells were lysed using 1X RIPA buffer, and protein concentrations were determined using the BCA assay (23225, Thermo). 1 μg/ul protein was mixed with 4X sample buffer with reducing agent and boiled at 95 °C for 10 min. Next, 20 μg of boiled protein was loaded onto a NuPAGE 4–12%, Bis-Tris gradient gel (#NP0335, Thermo) and samples were run at 130 V for about 1.5 h. Proteins were transferred to nitrocellulose membrane using iBlot dry transfer system (Program 4/10 min). Next, membranes were blocked using 5% milk dissolved in TBS-T (20 mM Tris, 150 mM NaCl, 0.1% Tween 20; pH 7.6) for 1 h, rocking at room temperature (RT). After blocking, membranes were incubated with primary antibody (1:1000 dilution) (Genetex CCDC170 antibody (#GTX107144), Cell Signaling Technology GAPDH (14C10) antibody (#2118) prepared in blocking buffer overnight at 4 °C. The next day, membranes were washed with TBS-T 3 times for 5 min. Then, they were incubated with secondary antibodies (1:10,000) (Anti-Rabbit IgG (H + L) (Promega,# W4011) diluted in blocking buffer for 1 h at RT. After the incubation, membranes were again washed 3 times for 10 min. Lastly, membranes were covered with western blot detection reagents (37074, Thermo Fisher) and visualized using the iBright imaging system.

## ChIP-Seq and qPCR

To cross link histones to DNA, formaldehyde (37%) was added to the culture media of a 150 mm × 25 mm plate (at -80% confluency) to a final concentration of 1% and incubated for 15 min at 37 °C with intermittent agitation. Cross-linking was quenched by adding 50 μL 2.5 M Glycine per 1 mL media and incubating for 5 min at 37 °C. Next, the plate was aspirated, removing as much media as possible and the cells were washed twice with ice cold PBS containing protease inhibitors (50 mL 1X PBS with 1 complete protease inhibitor tablet (Thermo Scientific, cat. # A32965) by adding 8 mL PBS/PI at a time. Aspirated PBS/PI and add 8 mL PBS/PI. The fixed cells were scraped using a plastic cell scraper and collected in a 50 mL conical tube. Cells were centrifuged at 4 °C for 10 min at 845 × g. The supernatant was next aspirated and discarded being careful not to disturb the cell pellet. The cell pellet was resuspended in SDS Lysis Buffer with 1X Halt™ Protease and Phosphatase Inhibitor Single-Use Cocktail (Thermo Scientific, cat. # 78442) (SDS Lysis Buffer: 1% SDS, 10 mM EDTA, 50 mM Tris-HCl, pH 8.1) at a ratio of 1 mL for every 2 × 10 cells. The cell pellet was next incubated on ice for 10 min. After redistributing the cell pellet into Bioruptor Pico 1.5 mL tubes (cat#C30010016) so that each tube contained no more than 200 μL, the cell pellet was sonicated (Bioruptor Pico sonication machine, cat. # B01080010, 10 cycles 30 s on/30 s off) to fragment sizes of 200−500 bp at 4 °C. The tubes were kept in a water/ice bath during sonication to prevent denaturation of the DNA. The tubes were then centrifuged at 24 °C for 10 min at max speed on an Eppendorf tabletop microcentrifuge. After centrifugation, the supernatant was collected from all the tubes and re-pooled into a 50 mL conical tube. ChIP Dilution Buffer (0.01% SDS, 1.1% Triton X-100, 1.2 mM EDTA, 16.7 mM Tris-HCl, pH 8.1, 167 mM NaCl) and 1X Halt™ Protease and Phosphatase Inhibitor Single-Use Cocktail (Thermo Scientific, cat. # 78442) were then added at a ratio of 5 mL buffer per 1 mL sonicated supernatant. The tube was inverted several times to ensure adequate mixing. Samples were always kept on ice during these steps. 250 μL of the diluted sample was collected as a whole cell extract (WCE) control and stored at 4 °C until the reverse cross-linking step, at which point 10 μL of 5 M NaCl, 25 μL 10% SDS and 1.25 μl 1 M DTT were added and the WCE was treated the same as the sample. 1 μL of antibody (Anti-Histone H3 acetyl K27 antibody - ChIP Grade, 1 mg/mL, Abcam, cat. #ab4729) per 1 mL buffer was added to the diluted sample and the sample was incubated overnight at 4 °C on a rotator. 15 μL of washed Dynabeads Protein G (Thermo Scientific, cat. # 10003D) or 15 μl Dynabeads Protein A (Thermo Scientific, cat. # 10001D) were mixed 30 μL of the slurry was added per sample. This mixture was allowed to incubate overnight at 4 °C on a rotator for 2 h. The Dynabeads were removed from solution using a magnetic tube holder and the supernatant was discarded. The beads were re-suspended with the DNA-histone-antibody complex in 1 mL of Low Salt Immune Complex Wash Buffer (0.1% SDS, 1% Triton X-100, 2 mM EDTA, 20 mM Tris-HCl, pH 8.1, 150 mM NaCl) and were incubated for 5 min at 4 °C in rotator. After incubation, the beads were again insolubilized using the magnet. These wash steps were performed twice each and repeated using 1 mL of LiCl Immune Complex Wash Buffer (0.25 M LiCl, 1% NP40, 1% deoxycholate, 1 mM EDTA, 10 mM Tris-HCl, pH 8.1) and then 1 mL of 1X TE Buffer pH 8.0 (10 mM Tris-HCl, 1 mM EDTA, pH 8.0). Finally, the bead pellet was re-suspended in 125 μl of elution buffer with DTT (Thermo Scientific, cat. # P2325) (1X TE, pH 8.0, 1% SDS, 150 mM NaCl, 5 mM DTT added just before use). The beads were allowed to elute at 65 °C for 10 min. The beads were then separated, and the supernatant was collected in a separate tube. The elution step was repeated once more and both elutions were pooled to a total volume of 25 μL. Cross-links reversed by incubating the sample at 65 °C overnight. The next day, the sample was allowed to cool to room temperature before adding Proteinase K (Thermo Scientific, cat. # EO0491) (5 μL Proteinase K, ~20 mg/ml and 245 μl

elution buffer) and incubating at 52°–55 °C for 2 h. After the Proteinase K treatment, nucleic acid was extracted from the sample using the phenol/chloroform extraction method and the concentration was quantified using a Qubit Fluorometer.

For library preparation, NEBNext Ultra™ II DNA Library Prep Kit for Illumina (cat. # E7645S), NEBNext® Multiplex Oligos for Illumina (cat. # E7335S) and AMPure XP beads (Beckman Coulter, cat. #A63881) were used according to manufacturer's instructions using 10–11 cycles of PCR amplification. Single-end sequencing of all ChIP libraries was performed on the 2 × 150 HiSeq. Quality of FastQ files was confirmed using FastQC (www.bioinformatics.babraham.ac.uk). Reads were then aligned to the GRCh38 human genome assembly using bowtie2 (2.4.1)[65] and sorted using samtools (v 1.6). Then Picard (2.21.4) (https://broadinstitute.github.io/picard/) was used for deduplication of the BAM\ files and Deeptools was used to convert the BAM files to Bigwig files (bamcoverage tool (RPGC normalized)).

### RNA seq and the analysis
Total RNA was isolated using the Zymo research Quick RNA miniprep kit utilizing the on-column DNAse treatment according to the manufacturer's instructions (R1054). The overall RNA purity was assessed by absorbance at 260 and 280 and the potential degradation was assessed by running on the agarose gel. Samples with (A260/A280 ratio ~2 and 28S and 18S band intensity ratio greater than 2 were accepted as pure and non-degraded and processed for qPCR and RNA-Seq analysis. RNA was prepared for sequencing using the NEBNext® Poly(A) mRNA Magnetic Isolation Module (NEBNext, E7490) and NEBNext® Ultra™ Directional RNA Library Prep Kit (NEBNext, E7420) according to the manufacturer's instructions. Paired-end sequencing of all RNA libraries was performed on the Illumina NextSeq 500 Platform.

The quality of FastQ files of the RNA seq data was checked using FastQC (www.bioinformatics.babraham.ac.uk). RNA-seq reads were aligned to the GRCh38 human genome assembly (Ensembl release 102) using the STAR aligner (v1.9.0) with default settings. BAM files deduplicated using Picard (v2.6.0) and were converted into bigwig files using bam coverage/DeepTools (v3.5.1) (bin size 1, normalized RPKM).

### Leiomyoma and myometrium RNA-Seq, H3K27ac and Hi-C data analysis
All leiomyoma and myometrium H3K27Ac-ChIP, HI-C-Seq and the Moyo et al. RNA-seq results were downloaded from NCBI-GEO data repository via accession GSE128242[26]. Paul et al. RNA-seq results were downloaded from NCBI-GEO data repository via accession GSE169255. Bigwig files were uploaded to the UCSC genome browser (GRCh38/hg38) for visualization. H3K27Ac-ChIP intensity scores around hg38-human-tss (-10 kb/+10 kb) were obtained using the DeepTools (computeMatrix tool)[66]. refTSS-human-hg38 (v3.1)[67] bed file and H3K27Ac-ChIP bigwig files (PT886, PT916, PT967, PT1063, PT848) of leiomyoma/myometrium were used with DeepTools. TSS-H3K27Ac intensity scores in genomic risk loci were obtained in R (cran.r-project.org) using Genomic Ranges (1.40.0). Differential intensity scores of leiomyoma/myometrium peaks were calculated using a two-sided student's t test in R.

HI-C-seq FAST-q files (PT886, PT916, PT967, PT1063) of leiomyoma/myometrium were downloaded and follow original publication data processing steps[26]. HI-C contact strength scores in genomic risk loci of leiomyoma/myometrium were obtained using the CHiCAGO pipeline[68] with default settings and differential contact strength scores were calculated using a two-sided student's t test in R. WashU genome browser[69] was used for visualization. The RNA-seq data of leiomyoma/myometrium (n = 15 patients) were downloaded and the original publication data processing steps were followed[26]. We analyzed the data using the cut-off values (FDR < 0.05). The RNA-seq datasets were analyzed in R using DESeq2 (version 1.30.1). The

heatmaps of differentially expressed genes were drawn in R using pheatmaps.

### RT-qPCR for gene expression of dCAS9-KRAB and dCas9-P300 human myometrial hTERT cells
RNA was isolated with TRIzol Reagent Protocol[67,70] (Invitrogen, cat # 15596018) and treated with DNase I (Nebnext, cat#M0303S). Next, the cDNA was converted with the High-Capacity RNA-to-cDNA™ Kit (Fisher scientific, cat# 4387406). We performed qPCR using the QuantStudio-3 (ThermoFisher) and Quanti Fast SYBR Green PCR Kit (Qiagen, cat # 204056), following manufacturer's instructions. 5 ng cDNA, 400 nM F and 400 nM R primers were used per reaction. Relative gene expression was calculated with the delta-delta CT method. Normalizations were made to the GAPDH gene as an endogenous control. For the CHIP-qPCR analysis, the ZNF333 gene was used as the negative control and the RPL10 gene was used as the positive control. Multiple unpaired student t-tests were used for the statistical analyses. All primer sequences are listed in Supplementary data 2.

### The single-cell RNA-seq data analysis
Single cell data of MyoF, and MED12 positive leiomyoma were downloaded from the GEO database under the accession number GSE162122. The data was then analyzed using the R package Seurat (version 4.3.0.1)[68,71]. Before the data integration step, quality control steps were performed. Cells with fewer than 200 or greater than 2500 feature counts, and more than 7% mitochondrial RNA content, >1% HBA1/HBA2/HBB (erythrocytes), and detectable EPCAM/KRT18 expression (endometrioid cells) were filtered out to remove low-quality cells. After the data was transformed using the SCTransform function of Seurat, the patient samples were integrated into a single object using FindIntegrationAnchors and IntegrateData functions according to their tissue types.

### Cell clustering, visualization, and cell-type annotation
Principal component analysis (PCA) was performed with 20 principal components (PCs, determined by Elbow plot), followed by Uniform Manifold Approximation and Projection (UMAP) using the same number of PCs. Next, the cells were clustered using the FindCluster function of Seurat with a resolution of 0.5. To identify the cell type of each cluster, first, the marker genes described originally in the Goad et al. publication were plotted using Seurat's FeaturePlot function. In order to confirm these cell types, marker genes of every cluster were identified using the Wilcoxon rank-sum test option of the FindAllMarkers function. The top markers for each cluster were next submitted to the Gene Ontology website Enrichr and the top cell type was identified by the Descartes 2021 cell type database[63,72]. Clusters were then manually annotated according to these criteria. Average expression of candidate genes was calculated using the FetchData function in Seurat, and average expression was plotted using pheatmaps and the scale = "row" argument in order to get z-scores by row. The column and row order of the heatmap was manually set. Lastly the top cell-type-specific genes were chosen and shown using both the calculated z-scores as well as the percentage of cells expressing.

### Reporting summary
Further information on research design is available in the Nature Portfolio Reporting Summary linked to this article.

## Data availability
All GWAS data analyzed in this manuscript are available in the references provided. Publicly available data sets were accessed via the Gene Expression Omnibus (GEO), accessions GSE128242, GSE169255, and GSE162122. The source data for all figures, when applicable, is provided in the Source Data file. Source data are provided with this paper.

## Code availability

All code used to generate figures can be accessed at https://github.com/AdliLab/Fibroid-GWAS-Manuscript. (https://doi.org/10.5281/zenodo.10412871).

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

## Acknowledgements
We thank all members of Adli lab for their critical insights and recommendations during this study, which was supported in part by a pilot award (PI: Adli) from Northwestern Uterine Leiomyoma Research Center (P50HD098580). We sincerely thank Prof. Mete Civelek at the University of Virginia for his critical manuscript reading, guidance, and recommendations.

## Author contributions
MA conceptualized the study and wrote the manuscript. KB performed all data analysis and wet lab experiments. AD performed the computational analysis, and FA, FSP and HE helped with the experiments.

## Competing interests
The authors declare no competing interests.
