## [Peer Review File · Nature Communications]

Integrating leiomyoma genetics, epigenomics, and single-cell transcriptomics reveals causal genetic variants, genes, and cell typesREVIEWER COMMENTS

Reviewer #1 (Remarks to the Author):

The manuscript by Buyukcelebi describes multiomics studies of uterine leiomyomas (ULs) using available data from other studies as well as their own. The goal is to leverage the huge GWAS data sets to identify and characterize new genes that could be important for UL biology for further analysis using expression and chromatin state data. Issues were noted however that should be addressed.

Major issues:

1. The ENCODE data used in figure 1, 5 is from smooth muscle cells or tissues? If not and these are from multi-cell type data, the authors should show results with smooth muscle cell data. This is a major limitation of the study that limits the physiological relevance of the study.
2. The authors use data from a custom promoter capture Hi-C, which biases their analyses. This custom chromatin capture is not described. This is another major limitation of the study that complicates interpretation of the results.
3. On line 166, the authors indicate that they were comparing the chromatin and GWAS data with differentially expressed genes, but they do not describe this DEG data, where it came from, or how they analyzed it.
4. On line 168, they indicate that 106 of the 394 risk loci target genes were differentially expressed in leiomyomas. Different leiomyoma/myometrial RNA-seq analyses have deposited their data. Are these 106 genes reproducibly differentially regulated. That is, the authors should validate these results in another data set.
5. The single cell RNA-seq results the authors show suggests that most of the cells are not smooth muscle or fibroblasts. This raises concerns about the physiological relevance of these data since myometrium and leiomyomas are predominantly composed of smooth muscle/fibroblast cells.
6. Where is the data showing SYNE1 is repressed in mesenchymal stem cells?
7. The authors should compare their GWAS results with another that used trans-ethnic meta-analyses (PMC6582231) to determine whether their risk loci are also found in that study. The Gallagher publication includes self-reported ULs in addition to clinically documented ULs, and only women with white European ancestry were part of the study.
8. Figures are not well prepared, e.g., see x axes in figure 1a, c, 2b #genes in loci don't line up correctly with the tick marks. Where is panel 3e? Legends do not describe the figures well, e.g., what is the dashed line in 1b, c? What does the y axis in 3a "Myometrium SC" mean? What is "heat" in box plots mean; Z-score, cpm? What do the boxes in 3b, c mean? There are more. These raise concerns about the rigor of the study.
9. ESR1 is not one of the genes identified in fig 2 as differentially expressed. What is the relevance for studying it here? Aren't these data contradictory? How does down-regulation of ESR1 expression related to chromatin interaction?
10. KANK1 "a well-established UF risk locus target genewhich [sic] is generally upregulated in UF tumors" on lines 198-199, is not shown upregulated in leiomyomas in fig 3c. Are these results contradictory and/or meaningful? Artifact?
11. On line 291, the authors indicated that the dCas9-KRAB mediated repression leads to "significant repression of the KANK1 itself." In the cell line, but the opposite is shown in fig 5c.
12. rs10815466 is associated with higher H3K27ac and HiC contacts in leiomyoma tissues in fig 2b. How does repression of H3K27ac and subsequent upregulation of KANK1 expression in fig 5c corroborate their analyses? Lines 298-299 also indicates "upregulation" of KANK1 is linked to the

lead SNP activation in leiomyomas, but they show activation with rs10815466 targeted repression in myometrial cell line. These results are confusing and do not support this statement.

Minor issues:

1. Previous publications are not well cited. For example, line 62-64 and 64-66.
2. What does "the FUMA analysis identified 394 genes as potential targets of the 34 lead SNPs" on line 133 mean? Genes are not target of SNPs.
3. line 258 should be figure 4c
4. Please explain what a "causal target of the risk loci that contributes to UF pathogenesis" on line 251 means. Also provide a reference.

Reviewer #2 (Remarks to the Author):

Buyukcebebi et al. re-analyzed publicly available GWAS data for uterine leiomyoma (UL). The authors then re-annotated the results, surveyed expression level of candidate genes in single-cell data and tested two lead candidates with CRISPR-KRAB epigenetic editing. While this study is a good example of how publicly available datasets can be used to assist interpretation of GWAS results, the significance appears to be limited in its current form. In particular, evidence for substantial conceptual innovation or advancement over existing publication is not clear, and the analyses presented lacked sufficient depth or interpretation.

Major

1. This paper is built upon a previously published GWAS study. The hits were reannotated with a published tool FUMA, with additional survey for expression levels in published single cell data, and eventually two candidate loci tested with established epigenetic editing. There appears to be a lack of substantial conceptual innovation or advancement over existing publication. It is also unclear how the reanalysis of the Gallagher data was done, to yield 1653 instead of 1172 SNPs.
2. Figure 3a appears to be reproduction of a previous published study (Goad et al), and it is unclear what conclusions can be drawn from Figure 3b-c. It is also unclear why there are so many gray cells (and what the gray color meant). Figure legend doesn't match the panels either. 'c-e' in the legend seems to be referring to Panel D.
3. Similarly, some other analyses presented lack depth or careful interpretation. For example, it is shown that 106 genes in Figure 2c are dysregulated out of 394 candidate genes. It would be helpful to know how does this compare to chance/background, and have the significance of this result spelled out.
4. The authors tested two candidate loci. However, neither experiments seemed to be conclusive with regard to the effect of induced silencing of these elements, and discussion/interpretation was minimal. In addition, the authors did not seek to establish the link between the alternative alleles and activity of these regulatory elements. More importantly, testing 2 out of 394 genes appears to also be more just a proof-of-principle.

Minor

1. The manuscript has some sloppy phrases, sentences, and spelling. E.g., 'Integrative analysis of uterine fibroid GWAS analysis with multi-omics data' in Figure 1 caption (= analysis of analysis?), 'Leiomyeme' in Figure 3d (Leiomyoma), 'after locus specific epigenetic (editing)' in Discussion, 'candite' (candidate), etc. It would help to improve readability of the manuscript if these are fixed.

Reviewer #3 (Remarks to the Author):

In this study, Kadir Buyukcebebi et al conducted integrative analysis including epigenetics, transcriptome, 3D chromatin structure, single-cell analysis, and GTEx to identify potential causative genes in uterine leiomyoma GWAS risk loci. In addition, they conducted locus-specific CRISPR-based epigenetic editing in two genetic loci. By integrating these multi-omics datasets,

they identified nearly 400 potential causative genes in 24 genetic region. The methods and result obtained in this study is valid and interesting, but I have several concerns with this paper as mentioned below

Major comments

1. The authors made a list of genes those are likely to be regulated by risk loci identified GWAS meta-analysis. The result of epigenetic editing also suggests causative genes in risk loci. However, there is no data that SNPs cause Epigenetic changes. Thus, the mechanism whereby SNPs in risk loci directly regulate the expression of UL-related genes is unclear. I considered that this study lacks novelty to be published in Nature communications.
2. In addition, roles of possible causative genes in UL pathogenesis are not clarified in this study. If SNPs in the SYNE1 region regulate CCDC170 not ESR1, what is the role of CCDC170 in UL? Functional analysis of CCDC170 in UL development is necessary.
3. Many pQTL studies have been published so far. The authors should include pQTL data in their integrative analysis.
4. Genome-wide expression analysis is necessary for epigenetic editing study not only for nearby genes, because QTL analyses could identify many trans-eQTL in addition to cis-QTL.

POINT BY POINT RESPONSE TO REVIEWER'S COMMENTS

Reviewer #1 (Remarks to the Author):

The manuscript by Buyukcelebi describes multiomics studies of uterine leiomyomas (ULs) using available data from other studies as well as their own. The goal is to leverage the huge GWAS data sets to identify and characterize new genes that could be important for UL biology for further analysis using expression and chromatin state data. Issues were noted however that should be addressed.

Major issues:

1. The ENCODE data used in figure 1, 5 is from smooth muscle cells or tissues? If not and these are from multi-cell type data, the authors should show results with smooth muscle cell data. This is a major limitation of the study that limits the physiological relevance of the study.

*Answer: The ENCODE data used in **Figure 1** is from all available cell lines and tissues that have been profiled as part of ENCODE and REMC projects. However, the data shown in **Figure 5** is from normal myometrium and uterine fibroid tissue, which is largely smooth muscle and fibroblast cells.*

*We do appreciate the reviewers' concern. The reason we used all ENCODE data in Figure 1 is because we wanted to incorporate as many regulatory elements and expressed genes in the genome as possible to identify any potential link to any genes to the risk loci in the genome. We partially agree with the reviewer that integrating GWAS data with smooth muscle cells makes better sense. However, although smooth muscle cells are the origin of uterine fibroids, we actually don't know the exact cell type through which GWAS risk loci work. We, therefore, in **Figure 3** present the scRNA-Seq analysis that has been performed in number of normal myometrium and uterine fibroid tissue. We highlight that several cell types in the fibroid tissue display unique and differential expression of GWAS risk loci targets. GWAS SNPs can be enriched in the regulatory elements of cell types that may not be obvious in the first analysis. For example, Farh et al (PMID: 25363779) showed that the majority of GWAS-identified risk SNPs for Type 1 diabetes, which is associated with aberrant pancreas function, are highly enriched in immune cell types. Although fibroids are not autoimmune diseases, we wanted to use the largest available data sets to identify potential gene targets that can be linked to the GWAS-identified risk SNPs.*

*However, the data in **Figure 5** is from myometrium and uterine fibroid tissue. Furthermore, the H3K27ac ChIP-Seq track is from an hTERT-immortalized uterine smooth muscle cell. To further complement these findings, we also provide chromatin state and high-resolution Hi-C data from patient samples.*

2. The authors use data from a custom promoter capture Hi-C, which biases their

analyses. This custom chromatin capture is not described. This is another major limitation of the study that complicates interpretation of the results.

Answer: We now have better explained the custom promoter capture Hi-C data, how it has been acquired, and what it signifies. We have used capture **Hi-C** data (in addition to ENCODE Hi-C data that is being utilized in the background of FUMA analysis in Figure 1) because this data was obtained in normal myometrium and matched uterine fibroid tissue. We acknowledge the limitation of promoter capture Hi-C data however we in the absence of high resolution Hi-C data from fibroid tissue samples, we think this data provide valuable insight into the 3D genome organization in fibroid tumor state.

3. On line 166, the authors indicate that they were comparing the chromatin and GWAS data with differentially expressed genes, but they do not describe this DEG data, where it came from, or how they analyzed it.

Answer: We thank the reviewer for this notice. We acquired the gene expression data from previously published data where RNA-Seq has been performed on 15 normal myometrium and 15 MED12 mutant uterine fibroid tissue (Moyo et al., PMID: 32094355). We analyzed the data using the previously published cut of values (FDR<0.05). This information has now been added to the Materials and Methods section and highlighted (Line 986).

4. On line 168, they indicate that 106 of the 394 risk loci target genes were differentially expressed in leiomyomas. Different leiomyoma/myometrial RNA-seq analyses have deposited their data. Are these 106 genes reproducibly differentially regulated. That is, the authors should validate these results in another data set.

Answer: These genes were reproducibly differentially regulated in the data released by Moyo et al., (PMID: 32094355), where they performed RNA-Seq in 15 normal myometrium and 15 UF tissue. We appreciate the suggestion by the reviewer. We, therefore, also integrated our GWAS target genes in an independent data set. We now present data in Figure 2 using two independent data sets (GSE128242 and GSE169255) obtained in normal matched fibroid tissues.

5. The single cell RNA-seq results the authors show suggests that most of the cells are not smooth muscle or fibroblasts. This raises concerns about the physiological relevance of these data since myometrium and leiomyomas are predominantly composed of smooth muscle/fibroblast cells.

Answer: We are sorry that our previous cell cluster annotations were not easy to locate SMC and fibroblasts because we had labeled individual sub-cell types. We have substantially updated our previous figure. We did this to make it easy to understand and also to accommodate some recommendations from other reviewers. We have also included a comparative analysis between normal myometrium and Leiomyoma state.

As can be seen in both Figure 3a (new) and Figure 3B (also new analysis), the majority of cells in leiomyoma, composing 50% of all fibroid tissue is smooth muscle cells.

6. Where is the data showing SYNE1 is repressed in mesenchymal stem cells?

***Answer:** SYNE1 is downregulated in bulk RNA-Seq data based on Moyo et al. (Figure 2f). in single cell RNA-Seq it was more challenging to demonstrate the cell type that downregulate SYNE1. Therefore, in the updated Figure 3, we are focusing on significantly upregulated genes in fibroid tissue and study specific cell types that express these genes. This analysis is more robust and the upregulated genes can be more reliably demonstrated in single cell data than the downregulated genes due to sparsity of the data. Therefore, we no longer make the claim about SYNE1 being downregulated in mesenchymal stem cells.*

7. The authors should compare their GWAS results with another that used trans-ethnic meta-analyses (PMC6582231) to determine whether their risk loci are also found in that study. The Gallagher publication includes self-reported ULs in addition to clinically documented ULs, and only women with white European ancestry were part of the study.

***Answer:** We appreciate this suggestion. UF displays a significant racial disparity as patients of African-American ancestry have earlier, more frequent, and larger UFs. Unfortunately, the Gallagher et al., study includes, for the most part, White/European ancestry population. Therefore, as suggested by the reviewer, we have also integrated our findings with multiple other GWAS studies from diverse ancestries. Unfortunately, despite our sincere efforts and even after arranging the meetings with the first authors of this study, we faced significant challenge in obtaining the summary statistics for the indicated trans-ethnic GWAS study.*

*However, we have managed to obtain GWAS data from two additional and independent GWAS performed in diverse population backgrounds. One of such GWAS is UK biobank data that has been performed diverse populations from multiple continents including African American samples and the other one was Japan biobank data from Japanese ancestry. The new integrative analysis is now presented as a new figure, **Figure 2g**.*

Significantly, this new analysis highlighted the importance of ancestry in GWAS data. We have also described new section in the main text (highlighted in lines 213-222). We would like sincerely thank the reviewer for this valuable and constructive critique.

8. Figures are not well prepared, e.g., see x axes in figure 1a, c, 2b #genes in loci don't line up correctly with the tick marks. Where is panel 3e? Legends do not describe the figures well, e.g., what is the dashed line in 1b, c? What does the y axis in 3a "Myometrium SC" mean? What is "heat" in box plots mean; Z-score, cpm? What do the

boxes in 3b, c mean? There are more. These raise concerns about the rigor of the study.

Answer: *We thank the reviewer for making us aware of these mistakes and are we sorry to miss them during figure preparation. In our revised manuscript, we have updated all figures and added new ones as recommended by this and other reviewers. Our new figures were prepared with better attention and care. We have also updated the figure legends to better describe each panel in the figures.*

9. ESR1 is not one of the genes identified in fig 2 as differentially expressed. What is the relevance for studying it here? Aren't these data contradictory? How does down-regulation of ESR1 expression related to chromatin interaction?

Answer: *Since ESR1 is the most proximal gene to GWAS risk loci with the leading SNP in 3'-end of SYNE1 gene. Since ESR1 encodes for a nuclear transcription factor that acts as receptor for estrogen hormone and estrogen hormone is implicated in the pathogenesis of fibroid tumors, ESR1 has been thought to be a putative target of GWAS risk loci.*

We agree with the reviewer that it is below the threshold of significance in Figure 2. However, we targeted ESR1 as well as the actual risk loci lead SNP region in Figure 4 because ESR1 is the most proximal gene to the risk loci and it has been implicated in the pathogenesis of uterine fibroids by several independent groups (PMIDs:27872195, 18701604, DOI: 10.1101/291237). Nevertheless, the data we are showing is to highly that our locus specific epigenetic editing approach works and leads to substantial gene repression at the target site.

In this Figure, we have now included new data (Figure 4e) to highlight that CCDC170 may be the major gene target of risk loci. The text explaining this new finding is included in the highlighted lines 333-352.

10. KANK1 “a well-established UF risk locus target gene which [sic] is generally upregulated in UF tumors” on lines 198-199, is not shown upregulated in leiomyomas in fig 3c. Are these results contradictory and/or meaningful? Artifact?

Answer: *Our results presented in Figure 2f shows that KANK1 is consistently upregulated in two different bulk RNA-Seq data sets. Furthermore, our updated figures in Figure 3C and Figure 3D shows that KANK1 is specifically upregulated in smooth muscle cells and not in other cell types in the fibroid tissue. Therefore, the data is not contradictory. However, we acknowledge that the previous single cell data was. Not easy to clearly see. We apologies for that and therefore, we provided a new analysis and figure in Figure 3 to better highlight this.*

11. On line 291, the authors indicated that the dCas9-KRAB mediated repression leads

to “significant repression of the KANK1 itself.” In the cell line, but the opposite is shown in fig 5c. rs10815466 is associated with higher H3K27ac and HiC contacts in leiomyoma tissues in fig 2b. How does repression of H3K27ac and subsequent upregulation of KANK1 expression in fig 5c corroborate their analyses? Lines 298-299 also indicates “upregulation” of KANK1 is linked to the lead SNP activation in leiomyomas, but they show activation with rs10815466 targeted repression in myometrial cell line. These results are confusing and do not support this statement.

*Answer: We are very sorry for the mistake of including the expression data for the wrong gene as KANK1 expression data. To make sure of our data, we have actually performed a completely new experiment to repeat our previous results and to make sure about the robustness of epigenetic editing data. The new data, presented in **Figure 5C**, shows that upon repression of the lead SNP target site, KANK1 is significantly downregulated along with two other genes; DMRT2 and DMRT3. The expression levels of two most distal genes did not change and other genes were slightly upregulated.*

We have updated the main text and the text describing and interpreting these results is highlighted in line 373-387

Minor issues:

1. Previous publications are not well cited. For example, line 62-64 and 64-66.

Answer: We are sorry for this. We have now revised the text and cited the proper references.

2. What does “the FUMA analysis identified 394 genes as potential targets of the 34 lead SNPs” on line 133 mean? Genes are not target of SNPs.

Answer: We have now included an expanded text to better describe the FUMA analysis and how these genes are linked to the GWAS target loci in highlighted lines 137-140.

3. line 258 should be figure 4c

Answer: We thank you for this notice, and we have corrected this mistake and the Figure is now updated.

4. Please explain what a “causal target of the risk loci that contributes to UF pathogenesis” on line 251 means. Also provide a reference.

Answer: We have updated the text to better explain this statement, and we have now provided appropriate references for the updated statement.

Reviewer #2 (Remarks to the Author):

Buyukcebi et al. re-analyzed publicly available GWAS data for uterine leiomyoma

(UL). The authors then re-annotated the results, surveyed expression level of candidate genes in single-cell data and tested two lead candidates with CRISPR-KRAB epigenetic editing.

While this study is a good example of how publicly available datasets can be used to assist interpretation of GWAS results, the significance appears to be limited in its current form. In particular, evidence for substantial conceptual innovation or advancement over existing publication is not clear, and the analyses presented lacked sufficient depth or interpretation.

Answer: We thank the reviewer for appreciating that our study is a good example of how publicly available datasets can be used to assist the interpretation of GWAS results. Indeed, this is one of the purposes of this study to demonstrate what can be done to further advance our understanding of GWAS risk loci.

Major

1. This paper is built upon a previously published GWAS study. The hits were reannotated with a published tool FUMA, with additional survey for expression levels in published single cell data, and eventually two candidate loci tested with established epigenetic editing. There appears to be a lack of substantial conceptual innovation or advancement over existing publication. It is also unclear how the reanalysis of the Gallagher data was done, to yield 1653 instead of 1172 SNPs.

Answer: We have now improved our paper by providing multiple panels of new data, such as the inclusion of serum protein levels as potential pQTL targets (Figure 2c) and additional gene expression data from independent resources (Figure 2f) and including multiple additional GWAS data set to highlight the importance of ancestry and population diversity in GWAS data (Figure 2g). Furthermore, we have re-analyzed the single cell data to better interpret it and make it more easily understandable by the readers (Figure a. Figure 3b and Figure 3c). Finally, we include a brand-new data (Figure 6) demonstrating a proof of principle that in certain GWAS loci, with limited genomic activity in the model systems, we can use locus specific epigenetic activation as an assay to target gene targets.

We believe that these improvements provide substantial depth. Furthermore, the analysis, the findings as well as overall research methodology will be of a significant use for UF researchers as well as a broader research community.

Regarding the critiques of how FUMA yielded higher number of significant SNPs, we believe there are two potential reasons. Firstly, FUMA incorporates 18 biological data repositories and tools to process GWAS summary statistics and provide a variety of annotations (Watanabe et al. Nature Com., 2017). Secondly, some of the data in the original Gallagher et al sample were not publicly available. We think that might be another reason for the discrepancy.

2. Figure 3a appears to be reproduction of a previous published study (Goad et al), and it is unclear what conclusions can be drawn from Figure 3b-c. It is also unclear why there are so many gray cells (and what the gray color meant). Figure legend doesn't match the panels either. 'c-e' in the legend seems to be referring to Panel D.

*Answer: In light of critiques, we have now reanalyze the entire Figure 3 to highlight the novelty of our analysis and re-visualize the findings to increase the understandability. Our new **Figure 3b** and **Figure 3c** are completely new analysis to highlight overall cell compositional changes in fibroid tumors vs. normal myometrium and to highlight the specific cell types that display high expression of GWAS-loci target genes.*

3. Similarly, some other analyses presented lack depth or careful interpretation. For example, it is shown that 106 genes in Figure 2c are dysregulated out of 394 candidate genes. It would be helpful to know how does this compare to chance/background, and have the significance of this result spelled out.

*Answer: We thank the reviewer for this suggestion. We have expanded the mentioned analysis by including an additional bulk RNA-Seq profiling data. We now have several new Figure panels (**Figure 2d**, **Figure 2e**, **Figure 2f**).*

We have also updated the main text in the manuscript to better describe this analysis and interpret the finding. The new section is highlighted text in lines :202-224. Regarding the significance of the overlap, we used Fisher exact test. We added the following text in the manuscript. "Performing a Fisher's exact test revealed that the interaction between the number of candidate gene targets and differentially expressed genes is very significant ($p = 6.235e-16$ with an odds ratio of 2.74 and $p < 2.2e-16$ with an odds ratio of 3.73, respectively). Of the differentially expressed candidate target genes, 87 were significantly upregulated ($FDR < 0.05$), and 81 of them were downregulated".

4. The authors tested two candidate loci. However, neither experiments seemed to be conclusive with regard to the effect of induced silencing of these elements, and discussion/interpretation was minimal. In addition, the authors did not seek to establish the link between the alternative alleles and activity of these regulatory elements. More importantly, testing 2 out of 394 genes appears to also be more just a proof-of-principle.

Answer: We thank the reviewers for this suggestion. We have taken several measures to improve the significance of our revised manuscript. Firstly, we have now included additional gene expression data from independent resources. Moreover, we now also integrate serum protein level data (of more than 4,000 proteins) as an additional quantitative trait. Furthermore, we have also included to additional GWAS studies to highlight the significance of population ancestry. Such integrative multi-omics analysis for uterine fibroid GWAS data is being done for the first time.

We have also significantly expanded the discussion and interpretation of our locus-specific epigenetic editing results.

*More critically, we are adding two completely new set of data for the functional characterization of GWAS target loci and reach a conclusive decision. We have performed new experiment to show that one of the gene targets that we linked directly to the epigenetic state of lead SNP in SYNE1 gene is a proximal gene called CCDC170. Through CRISPR-based genetic depletion strategy, we now have a new functional data that this gene is required for proliferation of SMC (**Figure 4e**).*

*Furthermore, we are now also adding that data where we used locus specific targeted epigenetic activation to map gene targets of the risk loci. This data is now presented in several panels in **Figure 6**. The significance of this figure is that we are demonstrating a proof of principle that GWAS risk loci can be targeted with not only CRISPR dCas9-KRAB based epigenetic repression strategy but also by dCas9-P300-based epigenetic activation if the target loci is not active in the cell line model.*

Altogether, we have functionally targeted 3 of the 24 GWAS risk loci in uterine fibroids.

Minor

1. The manuscript has some sloppy phrases, sentences, and spelling. E.g., 'Integrative analysis of uterine fibroid GWAS analysis with multi-omics data' in Figure 1 caption (= analysis of analysis?), 'Leiomyeme' in Figure 3d (Leiomyoma), 'after locus specific epigenetic (editing)' in Discussion, 'candite' (candidate), etc. It would help to improve readability of the manuscript if these are fixed.

***Answer:** We are very thankful for highlighting these mistakes. We have now performed an extensive editorial review and corrected the indicated and additional typos throughout the manuscript.*

Reviewer #3 (Remarks to the Author):

In this study, Kadir Buyukcelebi et al conducted integrative analysis including epigenetics, transcriptome, 3D chromatin structure, single-cell analysis, and GTEx to identify potential causative genes in uterine leiomyoma GWAS risk loci. In addition, they conducted locus-specific CRISPR-based epigenetic editing in two genetic loci. By integrating these multi-omics datasets, they identified nearly 400 potential causative genes in 24 genetic region. The methods and result obtained in this study is valid and interesting, but I have several concerns with this paper as mentioned below

***Answer:** We deeply appreciate that the reviewer sees our findings from this study as valid and interesting. We have now performed several additional experiments and data*

analyses to improve the significance of our manuscript and address the remaining concerns which we are summarizing our answers below.

Major comments

1. The authors made a list of genes those are likely to be regulated by risk loci identified GWAS meta-analysis. The result of epigenetic editing also suggests causative genes in risk loci. However, there is no data that SNPs cause Epigenetic changes. Thus, the mechanism whereby SNPs in risk loci directly regulate the expression of UL-related genes is unclear. I considered that this study lacks novelty to be published in Nature communications.

***Answer:** We appreciate this comment. More than 90% of disease associated variant SNPs are located in the non-coding region in the genome (Maurano et al. Science, 2012). Identifying the gene targets of lead SNPs is challenging because each lead SNP can be linked to multiple putative target genes due to genomic proximity, 3D interaction or being linked to the alternative allele through the expression level (eQTL).*

We agree with the reviewer that we did not show that a SNP is changing the epigenetic state of the risk loci. However, for all non-coding SNPs, the most plausible explanation and assumption is that the SNP in the regulatory element alters the transcription factor binding affinity and by doing so they alter the epigenetic state and 3D organization of the locus because differential binding of TFs then recruit additional cofactors.

*As reviewer highlighted, we did not show that a single nucleotide change can actually alter the expression of distal genes in this study. Such strategy requires precise genome editing and careful selection and study of large number of clones as we have done previously for a recurrent point mutation (Xiaolong et al, Genome Biology, 2019; Buyukcelebi et al., Nature Communications, 2023). However, in this study, we focused on larger scale perturbation using locus specific epigenetic editing to either repress two of the active loci or activate one of the loci and study the target genes. In **Figure 6**, we now provide a new experimental data to highlight the utility of the epigenetic activation system.*

2. In addition, roles of possible causative genes in UL pathogenesis are not clarified in this study. If SNPs in the SYNE1 region regulate CCDC170 not ESR1, what is the role of CCDC170 in UL? Functional analysis of CCDC170 in UL development is necessary.

***Answer:** We are thankful for this comment. Indeed, our new analysis and literature reading further highlight that CCDC170 could be critical target of this risk loci. We are now discussing this possibility in detail in the main text, highlighted lines: 333-352.*

This encouraged us to experimentally test the significance of CCD170 in uterine smooth muscle cells. We therefore used CRISPR to KO this gene at the genetic level and study how it effects survival and proliferation of SMC. Notably, our long-term live cell imaging

study, presented in Figure 4e, indicate that this gene is a critical cell survival and proliferation gene for uterine smooth muscle cells. Critically, these findings are in line with cells of breast cancer, where same loci is also identified as critical GWAS risk loci. We are discussing these findings in the main text, highlighted lines: 332-352.

3. Many pQTL studies have been published so far. The authors should include pQTL data in their integrative analysis.

Answer: We are very thankful for this constructive comment. We therefore performed new analysis by integrating our risk loci's with one of the largest and publicly available pQTL study. Specifically, we have acquired the data from Feringstad et al. (Nature Genetics, 2021). They have measured 4,907 proteins in serum from 35,559 individuals and associated each protein with the SNPs in the genome and identified the proteins that acts as a quantitative trait loci. Notably, we identified that 5 of our candidate UF lead SNPs are within close linear proximity to the genes coding for these pQTLs. Interestingly, some of the lead variants, such as rs78378222 is associated with 7 different pQTLs. These new analyses are now presented in Figure 2c and the text interpreting them is highlighted in lines: 171-197

4. Genome-wide expression analysis is necessary for epigenetic editing study not only for nearby genes, because QTL analyses could identify many trans-eQTL in addition to cis-QTL.

Answer: We totally agree with the reviewer that GWAS SNPs can be associated with trans eQTL genes as well. As suggested by the reviewer, we did perform RNA-Seq for both dCas9-KRAB repression and dCas9-P300 activation experiments and we did see the expected gene repression or activation at the target loci. These data are now presented as new **Figure 5c** and **Figure 6c** in the main Figures.

However, we find it challenging to claim and link the repression or activation of the target loci to changes in other genes in the genome. For example, for dCas9-KRAB repression experiment, we see ~200 differentially expressed genes (surprisingly majority of them are being upregulated) as shown in Figure below.

However, we are not sure if these differential expressed genes are directly linked to the epigenetic state of the KANK1 locus. We feel like these results could be due to secondary effects, for example, it could be due to the fact that one of the transcriptional repressor has been inactivated by repressing the target loci, which then led to activation of these genes. Because of such uncertainties, we did not feel making any claims and include this analysis in the main text. We think such claims are beyond the scope of this paper. We only included the expression status of the target region to highlight that our epigenetic repression and activation works.

REVIEWER COMMENTS

Reviewer #2 (Remarks to the Author):

In this revised manuscript, the research team have incorporated substantial new content that did help to improve the manuscript. However, there are still some concerns to be addressed, listed below.

1. Integration with different data modalities still appears to be quite superficial. For example, after the lead SNP (for the ESR1 region) is linked to CCDC170 with epigenetic editing, it would make sense to check in the eQTL data to check if the risk allele is indeed associated with higher expression of CCDC170. There is also no consideration of risk alleles overall, for other loci as well. In addition, after the discussion on potential cell type targets based on the expression of target genes, the CRISPR editing experiments were all in the same cell type, without showing if the myometrial cells were actually the potential cellular targets (e.g., is CCDC170 expressed in normal myometrium, etc.).

2. The CRISPR editing experiments to search for the gene target for each lead SNP are somewhat flawed as well. Aside from the cell type issue discussed above, there are several other things to address relevant to this point:

a. According to the H3K27Ac data shown in Figure 4a, hTERT-HM looks quite different in this region. Particularly, the big enhancer peak in the 3rd exon of ESR1 is missing. This might mean the results obtained in this cell line (in terms of regulatory circuits) may be different from what happens in normal myometrium and leiomyoma. Therefore, the conclusion about ESR1 being not the target of this particular SNP in leiomyoma development may not be true.

b. It is a bit unclear which region of the two highlighted loci is sgRNA1 target and which is sgRNA2 target. In addition, there does not seem to be contact between either the sgRNA1 target region or the sgRNA1 target region to the CCDC170 promoter, which does not make immediate sense if CCDC170 is the target.

c. It is not clear why only sgRNA1 data are shown (and not sgRNA2) beyond Panel B for Figure 5.

3. In the scRNA-seq analysis, row-normalized z scores are shown. However, such relative expression is not a good criterion to determine what the potential cellular target might be. Cell Type A and Cell Type B can both express a gene, while A has it lower than B, but A can still be the target (since the gene is still expressed in this data type).

Minor:

1. In Fig. 6c, the y axis scale is missing for each RNA-seq track (it is hard to judge if these tracks had the same y axis limits). Similar tissues for some of the other tracks in other figures as well.

2. In Figure 2e, making different parts of the Venn diagram proportional to the numbers might be more helpful; also, is there more overlap between FUMA mapped genes and the DEGs than expected by chance?

Reviewer #3 (Remarks to the Author):

The authors added many experiments, data in the revised manuscript, and fully addressed the reviewers' comments. This paper clarifies one aspect of the pathogenesis of leiomyoma, and I recommend this paper to be published in Nature Communications.

Reviewer #4 (Remarks to the Author):

Authors

The submitted revised manuscript combined multiomic studies to identify fibroid risk loci and their target genes. This reviewer appreciates that the authors used publicly available datasets as well as their own. Some comments have been addressed, but major concerns remain for the analysis, particularly the single cell RNA-seq, and interpretation of the results. Please see individual comments below.

Major comments

1- The author used several omic studies and compared fibroid tumors to "normal" myometrium. However, the origin of the control tissue has not been properly defined in some comparison sections. Please refer to MyoF for myometrium from a fibroid patient, or MyoN for myometrium from a non-fibroid patient. The use of MyoN or MyoF as a control may change the results (doi: 10.3390/ijms22073618). Similarly for the fibroid subtypes, it is known that each has different transcriptomic and methylation profiles raising the importance of clearly specifying the type of fibroid tissue use for each comparison.

2- The analysis of the single cell RNA-seq is inconsistent with the published study and, importantly, with respect to the myometrium cell composition. Since these results have been re-analyzed, it is important to detail the workflow and analysis. How many cells passed the quality control, to which version of the genome the cells were mapped and justifying why 30 PCs were chosen. The number of cells in the myometrium seems very low compared to the fibroid samples; the original study reported over 10,000 cells. Moreover, the author mentioned that they used published marker genes for defining cell clusters. Marker genes need to be listed and should be plotted to help the reader, for example, with a dotplot or feature plot. One major concern is the identification of epithelial cells in myometrium and fibroid single cell results. The original study and other single cell data from myometrium did not report the presence of epithelial cells. Histologically, epithelial cells are not present in the myometrium. The authors report in line 255 "Significantly, we found that SMC, the cell type of origin for UF tumors, contains the highest number of these highly expressed GWAS target genes in the leiomyoma samples. Nearly 44% of GWAS-targets genes upregulated in leiomyoma (39/87) are highly differentially expressed in the smooth muscle cell cluster (Figure 3c)". However, this result is not surprising since more than 50% of the cells are labeled by the authors as smooth muscle cells. The cell clusters need to be more specifically defined.

3- The authors reported that edited cells with a sgRNA targeting CCDC170 showed decreased proliferation and cell survival. The cell survival assay is not shown in figure 4, please add the corresponding figure/data. If cell survival decreased with sgRNA, it could affect the proliferation results by appearing as a false decrease.

Minor comments

1- Be consistent with abbreviations, UF was used for uterine fibroids in the abstract and switched to UFs for the rest of the manuscript. Since UF has been defined, please use UF throughout the text.

2- Overall, the paper is very dense and, at times, difficult to follow. This would be improved by removing discussion and methods statements from the results section. A few examples: lines 122-128 (methods), lines 169-170 (discussion), and lines 192-197 (discussion).

3- Lines 426-429: Nomenclature for the fibroid subtypes (MED12, HMGA2...) was already defined in the introduction section.

4- Please remove repeated text from the results, for example lines 121-122 and line 129. This makes the manuscript more difficult to read.

5- Effort from the authors to mention different fibroid subtypes, including rare subtypes like FH, is appreciated. However, the authors did not mention the HMGA1 subtype which represents 3% of fibroids (<https://doi.org/10.1038/s41586-021-03747-1>). – Please address

6- Typo line 141: "candidate"

7- eQTL: Please define for the first use.

8- Figure 4f: Typo CCDC170, not CCD170

9- Line 416: Replace qRT-PCR by RT-qPCR.

10- Which reference genes were used for qPCR? Please specify in the methods section.

11- Please avoid references to figures in the discussion section.

POINT BY POINT RESPONSE TO THE REVIEWER'S COMMENTS

Reviewer #2 (Remarks to the Author):

In this revised manuscript, the research team have incorporated substantial new content that did help to improve the manuscript. However, there are still some concerns to be addressed, listed below.

1. Integration with different data modalities still appears to be quite superficial. For example, after the lead SNP (for the ESR1 region) is linked to CCDC170 with epigenetic editing, it would make sense to check in the eQTL data to check if the risk allele is indeed associated with higher expression of CCDC170. There is also no consideration of risk alleles overall, for other loci as well. In addition, after the discussion on potential cell type targets based on the expression of target genes, the CRISPR editing experiments were all in the same cell type, without showing if the myometrial cells were actually the potential cellular targets (e.g., is CCDC170 expressed in normal myometrium, etc.).

Answer: We checked the risk allele and its correlation with the CCDC170 gene. CCDC170 is not an eQTL gene. However, it did appear in our FUMA analysis as a gene target of the GWAS loci due to 3D chromatin interaction. Furthermore, the SNP that we target in this manuscript (among others) has previously been shown to have contact with CCDC170.

It should be noted that recent systematic analyses by Mostafavi et al. (Nature Genetics, 2023) highlight that quantitative trait loci (eQTLs) explain only a small fraction of GWAS signals and that there is significant systematic differences between GWAS and eQTL variants. For example, they noted that “one analysis by the GTEx Consortium found that only 43% of GWAS hits (median 21% of hits per trait) were colocalized with eQTLs. Similarly, averaged across traits, only 11% of heritability is estimated to be mediated by gene expression in GTEx tissues”. This analysis highlights that the majority of the GWAS targets may not be an eQTL. In this study, we focused on the GWAS loci and identified putative target genes that may be regulated by the activity of the loci. As the reviewer noted, we did not focus on specific risk variants and risk alleles.

We agree with the reviewer regarding the limitation of using a single cell line in this study. As noted in our discussion, we highlighted this as a limitation of our study. Unfortunately, finding an appropriate cell line that recapitulates the in vivo tissue regulatory network is extremely challenging, if not impossible. In the case of fibroid tumors, no one could generate a cell line model for this disease as the mutant cells do not survive in 2D culture conditions.

2. The CRISPR editing experiments to search for the gene target for each lead SNP are somewhat flawed as well. Aside from the cell type issue discussed above, there are several other things to address relevant to this point:

a. According to the H3K27Ac data shown in Figure 4a, hTERT-HM looks quite different in this region. Particularly, the big enhancer peak in the 3rd exon of ESR1 is missing. This might mean the results obtained in this cell line (in terms of regulatory circuits) may be different from what happens in normal myometrium and leiomyoma. Therefore, the conclusion about ESR1 being not the target of this particular SNP in leiomyoma development may not be true.

Answer: This criticism is related to the above limitations of in vitro cell lines. We completely agree on the limitations of a cell line model. We had already highlighted this as a weakness of such perturbation studies in in vitro models. We have also revised the tone of our discussion that ESR1 is not a target, at least in this cell line, as discussed in lines 307-314 and 418-424.

b. It is a bit unclear which region of the two highlighted loci is sgRNA1 target and which is sgRNA2 target. In addition, there does not seem to be contact between either the sgRNA1 target region or the sgRNA2 target region to the CCDC170 promoter, which does not make immediate sense if CCDC170 is the target.

Answer: We agree and we like to emphasize that the Hi-C track signal is correlated with sequencing depth, and the current display is only showing the most robust 3D interactions for visual clarity. Indeed, even at the current depth of sequencing, there is a connection between this region and the CCDC170 gene region. A higher resolution Hi-C would have detected a more robust signal between the sgRNA1 region and the CCDC170 gene. We have focused on the loci and we observed substantial change in this gene when either sgRNA1 or sgRNA2 is used.

We are sorry that in Figure 4, the sgRNA targeting sites were shown with broad bands. We have edited the figure to highlight that both of the SNP targeting sgRNAs are targeting the same region very proximal to each other (200 bp apart). We have updated the figure to highlight this and we hope the new updated figure is clearer.

c. It is not clear why only sgRNA1 data are shown (and not sgRNA2) beyond Panel B for Figure 5.

*Answer: We observe stronger repression with sgRNA1 and thus, we decided to display the perturbation data with sgRNA1 only. However, we have also performed the experiment with sgRNAs as well. The updated version of the manuscript now contains this data as a new figure in **Supplementary Figure 5**.*

3. In the scRNA-seq analysis, row-normalized z scores are shown. However, such relative expression is not a good criterion to determine what the potential cellular target might be. Cell Type A and Cell Type B can both express a gene, while A has it lower than B, but A can still be the target (since the gene is still expressed in this data type).

Answer: *In theory, we agree with the reviewer's reasoning. We can not absolutely rule out that such genes are only expressed in the indicated cell types and have only an impact on those cells. We therefore performed additional analysis, such as quantifying the percent of cell types in each cluster expressing detectable levels of these genes. This new analysis, shown in **Supplementary Figure 3**, takes care of the cell number in each cluster as a confounding factor.*

We show both z-scores as well as the percentage of cells expressing a number of the FUMA-identified GWAS target genes. Looking at these values, it is clear that genes such as C1QA and C1QC are not just overexpressed in myeloid cell types, but are also expressed in nearly 50% of those cells. Conversely, these genes are expressed very lowly and are only detected in around 1.2% of SMCs. We argue that such a low amount of expression in this cell type is a strong indication that the genes are more likely to have a functional impact in myeloid cells.

Minor:

1. In Fig. 6c, the y axis scale is missing for each RNA-seq track (it is hard to judge if these tracks had the same y axis limits). Similar tissues for some of the other tracks in other figures as well.

Answer: *We thank the reviewer for this constructive critique. The scales are provided in the updated figures. The bigwig files for the RNA-seq data in Figure 6 were RPKM normalized. They are set to the same scale and the scale is now shown.*

2. In Figure 2e, making different parts of the Venn diagram proportional to the numbers might be more helpful; also, is there more overlap between FUMA mapped genes and the DEGs than expected by chance?

Answer: *Since the numbers are too different from each other, the display will not be as good if we do as suggested. We, therefore, prefer to display data as is, as the point is not to highlight the size differences.*

For the significance of overlap, the p-values were previously requested and supplied in the last revision. They were calculated using a Fisher's exact test and were found to be $<2.2 \times 10^{-16}$ for the Moyo et al. dataset and 6.235×10^{-16} for the Paul et al. dataset.

Reviewer #3 (Remarks to the Author):

The authors added many experiments, data in the revised manuscript, and fully addressed the reviewers' comments. This paper clarifies one aspect of the pathogenesis of leiomyoma, and I recommend this paper to be published in Nature Communications.

Answer: We appreciate that the reviewer is fully satisfied with our improvements.

Reviewer #4 (Remarks to the Author):

Authors

The submitted revised manuscript combined multiomic studies to identify fibroid risk loci and their target genes. This reviewer appreciates that the authors used publicly available datasets as well as their own. Some comments have been addressed, but major concerns remain for the analysis, particularly the single cell RNA-seq, and interpretation of the results. Please see individual comments below.

Major comments

1- The author used several omic studies and compared fibroid tumors to “normal” myometrium. However, the origin of the control tissue has not been properly defined in some comparison sections. Please refer to MyoF for myometrium from a fibroid patient, or MyoN for myometrium from a non-fibroid patient. The use of MyoN or MyoF as a control may change the results (doi: 10.3390/ijms22073618). Similarly for the fibroid subtypes, it is known that each has different transcriptomic and methylation profiles raising the importance of clearly specifying the type of fibroid tissue use for each comparison.

Answer: We appreciate the comment and this highlight. For both bulk RNA-seq datasets, only matched myometrium-leiomyoma samples were used (i.e. MyoF controls). For the single-cell dataset, all of the myometrium samples were obtained from patients who had confirmed fibroids (MyoF). We have now updated the text to highlight this.

2- The analysis of the single cell RNA-seq is inconsistent with the published study and, importantly, with respect to the myometrium cell composition. Since these results have been re-analyzed, it is important to detail the workflow and analysis. How many cells passed the quality control, to which version of the genome the cells were mapped and justifying why 30 PCs were chosen. The number of cells in the myometrium seems very low compared to the fibroid samples; the original study reported over 10,000 cells. Moreover, the author mentioned that they used published marker genes for defining cell clusters. Marker genes need to be listed and should be plotted to help the reader, for example, with a dotplot or feature plot. One major concern is the identification of epithelial cells in myometrium and fibroid single cell results. The original study and other single cell data from myometrium did not report the presence of epithelial cells. Histologically, epithelial cells are not present in the myometrium.

Answer: We appreciate this constructive critique. We have revisited our single-cell analysis pipeline to understand the differences between our analyses and the analyses in the original study. During this assessment, we have noted several differences between our pipeline and the pipeline used in the original study.

First of all, due to spelling error in the original files deposited by the authors, one of the myometrium samples was not included in our original analysis. This has been rectified for our current analysis.

More importantly, the original paper filtered out several cell types from their analysis. They computationally removed any cell types that are EPCAM(+) and KRT18(+) cells). This is the major difference between our analyses and theirs because they computationally removed all the epithelial cells from the downstream analysis where we had included them.

Furthermore, and in addition to the above difference, the number of cells that has been deposited into the database is significantly different than what has been reported in the published study. The datasets downloaded from the Gene Expression Omnibus for the original study, there are a total of 11807 cells present in the myometrium samples and 156810 cells in the leiomyoma samples before QC. Using the same thresholding criteria as the original paper (>200 and <2500 features, <7% mitochondrial reads, <1% HBA1/HBA2/HBB, and removing EPCAM(+) and KRT18(+) cells), we obtained 6394 myometrial cells and 41864 leiomyoma cells. These values are now reflected in the methods section. These numbers are substantially different than what has been reported in the original study. We can only assume that the authors have acquired additional data after they published their study and decided the upload all of it into the database. But this is a significant challenge when trying to replicate the exact same figures from the original study.

Nevertheless, we used the same QC criteria used in the original study and we have also removed EPCAM(+) and KRT18(+) cells. Using an Elbow Plot, we decided that 20 PCs is sufficient to explain the majority of the variation seen. Thus, the datasets were analyzed using these parameters. The full code used to create the data is uploaded to the Adli lab github repository: github.com/adlilab and the methods section has been updated.

*We, therefore, completely updated **Figure 3**. The new figure is perfectly in line with our previous Figure and still fully supports our previous conclusions.*

The authors report in line 255 "Significantly, we found that SMC, the cell type of origin for UF tumors, contains the highest number of these highly expressed GWAS target genes in the leiomyoma samples. Nearly 44% of GWAS-targets genes upregulated in leiomyoma (39/87) are highly differentially expressed in the smooth muscle cell cluster (Figure 3c)". However, this result is not surprising since more than 50% of the cells are labeled by the authors as smooth muscle cells. The cell clusters need to be more specifically defined.

***Answer:** Although we understand the reviewer's reasoning, we like to emphasize that the size of a cluster does not necessarily denote the number of differentially expressed genes within said cluster. For example, while endothelial cells represent a significant*

portion of both myometrium and leiomyoma tissue samples (around 42 and 17% respectively), they are barely represented in the list of differentially expressed GWAS targets compared to other, much smaller cell populations, such as myeloid cells (Fig. 3c). We therefore contend that the large number of SMCs is not a guarantee that the majority of differentially expressed genes would be in that cell population.

Furthermore, we now also provide additional analysis (**Supplementary Figure 3**) where we highlight the Z-score as well as the percent of cells in each cluster expressing the indicated genes. This analysis removes the total cell number as a confounding factor and highlights that the genes that are expressed highly in a specific cluster are also expressed in the majority of the cells in that cluster. For example, the data in Supplementary Figure 3 shows that genes such as C1QA and C1QC are not just overexpressed in myeloid cell types, but are also expressed in nearly 50% of those cells. Conversely, these genes are expressed very lowly and are only detected in around 1.2% of SMCs.

3- The authors reported that edited cells with a sgRNA targeting CCDC170 showed decreased proliferation and cell survival. The cell survival assay is not shown in figure 4, please add the corresponding figure/data. If cell survival decreased with sgRNA, it could affect the proliferation results by appearing as a false decrease.

Answer: We agree with this comment. We therefore provide a new data showing relative levels of cell death as measured by Apoptosis using the Incucyte live cell imaging platform. The new data shows that depletion of CCDC170 does not impact cell death, we, therefore, revised the text to highlight that depletion of CCDC170 impacts cell proliferation.

Minor comments1- Be consistent with abbreviations, UF was used for uterine fibroids in the abstract and switched to UFs for the rest of the manuscript. Since UF has been defined, please use UF throughout the text.

Answer: Thank you for your comments. We have revised the manuscript thoroughly according to your comments.

2- Overall, the paper is very dense and, at times, difficult to follow. This would be improved by removing discussion and methods statements from the results section A few examples: lines 122-128 (methods), lines 169-170 (discussion), and lines 192-197 (discussion)

Answer: Thank you for this comment. We have revised the text to make the text less dense.

3- Lines 426-429: Nomenclature for the fibroid subtypes (MED12, HMGA2...) was already defined in the introduction section.

Answer: Thank you.

4- Please remove repeated text from the results, for example lines 121-122 and line 129. This makes the manuscript more difficult to read.

Answer: Thank you. We have revised the text accordingly to the best of our ability.

5- Effort from the authors to mention different fibroid subtypes, including rare subtypes like FH, is appreciated. However, the authors did not mention the HMGA1 subtype which represents 3% of fibroids (<https://doi.org/10.1038/s41586-021-03747-1>). – Please address

Answer: Thank you, we have revised the text to highlight this.

6- Typo line 141: “candidate”

Answer: Thank you, it is done.

7- eQTL: Please define for the first use.

Answer: Thank you, it is done.

8- Figure 4f: Typo CCDC170, not CCD170

Answer: Thank you, it is done.

9- Line 416: Replace qRT-PCR by RT-qPCR.

Answer: Thank you, it is done.

10- Which reference genes were used for qPCR? Please specify in the methods section.

Answer: Thank you, we note that we used GAPDH.

11- Please avoid references to figures in the discussion section.)

Answer: Thank you, it is done.

REVIEWERS' COMMENTS

Reviewer #2 (Remarks to the Author):

The authors have addressed my previous concerns.

Reviewer #4 (Remarks to the Author):

The submitted revised manuscript combined multiomic studies to identify fibroid risk loci and their target genes. This reviewer appreciates that the authors answered comments and concerns. The paper after minor revision will be suitable for publication.

Minor comments

Please change "Normal" myometrium by MyoF or myometrium from a fibroid patient. It is not a normal myometrium. Line: 361, 398, 428, 502, 505, 515, 518, 520, 523, 525, 536, 552, 568, 997, and Figure 3 a.